**Cite this article:** von Mohr M, Kirsch LP, Fotopoulou A. 2021 Social touch deprivation during COVID-19: effects on psychological wellbeing and craving interpersonal touch. *R. Soc. Open Sci.* **8**: 210287.

Subject Areas:
behaviour/cognition/psychology

Keywords:
social touch, COVID-19, anxiety, wellbeing, attachment

Authors for correspondence:
Mariana von Mohr
e-mail: mariana.vonmohr@rhul.ac.uk
Louise P. Kirsch
e-mail: louise.kirsch@parisdescartes.fr

†These authors share first authorship.

# Social touch deprivation during COVID-19: effects on psychological wellbeing and craving interpersonal touch

Mariana von Mohr[1,2,†], Louise P. Kirsch[3,4,†] and Aikaterini Fotopoulou[5]

[1]Lab of Action and Body, Department of Psychology, Royal Holloway, University of London, London, UK
[2]Departamento de Psicología, Universidad Iberoamericana, México, Mexico
[3]Institute for Intelligent Systems and Robotics (ISIR), Sorbonne Université, Paris, France
[4]Université de Paris, INCC UMR 8002, 75006, Paris, France
[5]Research Department of Clinical, Educational and Health Psychology, University College London, London, UK

MVM, 0000-0003-0671-0735; LPK, 0000-0002-8418-776X

Social touch has positive effects on social affiliation and stress alleviation. However, its ubiquitous presence in human life does not allow the study of social touch deprivation 'in the wild'. Nevertheless, COVID-19-related restrictions such as social distancing allowed the systematic study of the degree to which social distancing affects tactile experiences and mental health. In this study, 1746 participants completed an online survey to examine intimate, friendly and professional touch experiences during COVID-19-related restrictions, their impact on mental health and the extent to which touch deprivation results in craving touch. We found that intimate touch deprivation during COVID-19-related restrictions is associated with higher anxiety and greater loneliness even though this type of touch is still the most experienced during the pandemic. Moreover, intimate touch is reported as the type of touch most craved during this period, thus being more prominent as the days practising social distancing increase. However, our results also show that the degree to which individuals crave touch during this period depends on individual differences in attachment style: the more anxiously attached, the more touch is craved; with the reverse pattern for avoidantly attached. These findings point to the important role of interpersonal and particularly intimate touch in times of distress and uncertainty.

# 1. Introduction

The COVID-19 pandemic presents a unique challenge to societies all over the globe. In order to hinder the accelerating growth of infections, changes in the core social habits of people have become essential. For example, citizens are required to engage in 'physical distancing', initially referred to as 'social distancing' by the World Health Organization (WHO); that is, the minimization of close contacts with others. Notably, the change in term was because it is important to encourage social interactions (e.g. virtual communications) during epidemic periods. Indeed, social connection and support, even in the form of texts [1], has beneficial effects on distressing events [2] and physical health [3,4]. In particular, social supportive behaviours following stress conditions seem to attenuate multiple stress systems, including the autonomic nervous system and hypothalamic–pituitary–adrenal (HPA) axis [4], possibly mediated by neuropeptides involved in social bonding and affiliative behaviour, including oxytocin [5]. Further, neuroimaging studies indicate that social support reduces activity in brain regions implicated in emotion regulation (i.e. anterior cingulate cortex, dorsolateral and ventrolateral prefrontal cortex) [6,7].

However, a particularly effective form of communicating (non-verbal) support, which in addition facilitates the formation and maintenance of social bonds, is touch [8–10]. The potential benefits of touch have been studied in many fields, ranging from animal studies to developmental and adult psychological and neuroscientific studies in humans [11–15]. On the one hand, social touch is thought to possess positive hedonic value (although clearly this value depends on the specific context, i.e. touch may not always be welcome or pleasant), in order to promote affiliative and prosocial behaviour [16]. For example, the effects of touch in social interactions have been shown to increase the liking of a person [17–19] as well as generosity and compliance [19–21]. On the other hand, social touch serves as a form of bonding and reinforcing alliances [22,23]. For example, in non-human mammals such as primates, grooming is typically observed within close conspecifics, such as in maternal behaviour, with neurotransmitters involved in social bonding (e.g. oxytocin) mediating such effects [24]. In humans, caregiving touch is essential for growth and development in infancy and for wellbeing and bonding in adulthood. Touch actively reduces infant stress by increasing positive affect [25,26] and calms infants in pain and discomfort [14]. In the context of attachment theory [27], studies support the facilitating role of touch in establishing the social bond between infant and carers [28–30]. Touch is most prevalent in close relationships [31] and also has a lifelong effect on human bonding. For instance, a recent study suggests that in romantic couples, self-reports of mutual grooming are positively correlated with relationship quality and previous experiences of familial affection [32].

Moreover, touch by conspecifics has analgesic and stress-alleviating effects [33] mediated by neurobiological pathways involved in social bonding [12]. In particular, in humans, social touch has been suggested as a stress buffer, playing a critical regulatory role in the body's responses, including cortisol and heart rate responses [34], to acute life stressors, which ultimately promotes social connection [35]. For example, a recent study suggests that touching a teddy bear mitigates feelings of social exclusion to increase prosocial behaviour [36]. Similarly, studies have shown that touch, such as a caress on the forearm and a rub on the back of the hand, reduces feelings of social exclusion [10] and the perception of loneliness [37], respectively. Moreover, functional neuroimaging studies have shown an attenuation of neural responses typically implicated in affective regulation when social embodied support (e.g. hand-holding by a romantic partner) is provided in the face of threat [6], including pain [38,39]. In line with the notion that we have adapted to the presence and active care of other conspecifics [40–42], our emotions and sense of selfhood are constituted on the basis of early social interactions, including touch [43,44]. As such, it is not surprising that touch deprivation is associated with negative outcomes. For example, in children, touch deprivation is associated with struggles in learning to speak [45], sleep problems and school performance [46] and aggression [47]. In adults, touch deprivation is associated with higher mood and anxiety symptoms [48], depression [49], perceived loneliness [37] and worse wellbeing more generally [15]. However, given physical distancing regulations during the COVID-19 pandemic, our ability to provide and receive this type of support has probably been affected. This stands to be particularly detrimental as the pandemic has signalled a period of global uncertainty, with various mental health consequences such as an increase in loneliness and symptoms of anxiety and depression [50].

Here, participants ($N = 1746$) were asked to complete an online survey to examine touch experiences during COVID-19, and whether these experiences are associated with individual's psychological wellbeing, specifically anxiety and feelings of loneliness. Specifically, we examined whether the amount of touch experienced in the past week (i.e. during COVID-19-related social restrictions) from

intimate, friendly or professional sources differently influenced the aforementioned measures of wellbeing. Given literature suggesting that touch is more powerful when provided by a close other [6,39] with its regulatory effects mediated by psychological intimacy [51], we expected that, in particular, the more intimate touch experienced in the past week, the better the wellbeing. By contrast, we expected that the more the lack of touch during COVID-19-related social restrictions (by taking into account touch experienced before COVID-19), the worse the wellbeing.

Furthermore, we also examined whether the degree to which participants would have wanted to experience intimate, friendly and professional touch in the past week depended on the number of days practising social distancing, controlling for lack of touch. Given literature suggesting that threats to social connection (e.g. ostracism or isolation) in turn promote seeking social reconnection and proximity [12,52,53], we expected that the more days participants had been practising social distancing, the more touch they would have wanted to experience in the past week.

In addition, it is known that different people crave touch to different degrees; in fact, some people may not even want to experience it in times of physical or social threat [54]. Therefore, we examined whether wanting touch in the past week (i.e. during COVID-19-related lockdown) depended on individual differences, such as (i) adult attachment style and (ii) attitudes and experiences towards touch. Adult attachment style was measured using a well-validated questionnaire (Experiences in close relationships—short; ECR-S) [55]. This questionnaire pertains to adult romantic relationships and takes a dimensional approach, yielding continuous scores of attachment anxiety and attachment avoidance. Given that attachment anxiety is characterized by a need for emotional closeness, worries about rejection and abandonment, and over-dependence on others, which seem to extend to touch perception [56], we expected that individuals scoring higher on attachment anxiety would report to have wanted to experience more touch in the past week. By contrast, given that attachment avoidance is characterized by a need for emotional distance, as well as a resistance to trusting and depending on others, we expected that individuals scoring higher on attachment avoidance would report to have wanted less touch in the past week.

With respect to positive attitudes and experiences towards touch, we adapted the Touch Experiences and Attitudes Questionnaire (TEAQ) [57] by selecting an item from each component (e.g. friends and family touch, current intimate touch, childhood touch, etc.). Here, we expected that individuals scoring higher on positive attitudes and experiences towards touch would report wanting more touch. Moreover, taking advantage of the TEAQ different components, we assessed whether individual items predicted wanting touch in the past week. We were particularly interested in the childhood touch component, given that attachment representations are thought to originate in early experiences with primary carer(s), and this may extend to touch behaviours [58,59]. In this sense, we expected that individuals scoring higher on childhood touch experiences would report wanting more touch in the past week.

# 2. Methods

## 2.1. Participants

As part of a larger study on touch during COVID-19-related social restrictions, 1746 participants were recruited online. This was a sample size of convenience based on a survey distributed as widely as possible within a given period of COVID-19-related restrictions. Participants were asked to complete a survey about their tactile experiences during (and before) COVID-19-related restrictions, as well as other self-reported measures about their wellbeing.

Seven hundred and forty-six participants were recruited via social media (e.g. Facebook, Twitter) and 1000 were recruited via Prolific (https://prolific.ac/) pre-screened for approval rating (i.e. how well participant performed in previous studies) at greater than 70% as well as for country of residence (UK, France and Mexico), as these countries had not yet lifted severe social distancing restrictions at the time of recruitment and were the countries that had the most prevalence in those recruited via social media. Participants recruited via Prolific were paid £2.50 for their time (approx. 20 min). Participants recruited via social media were recruited between the 21 April and the 13 May 2020; participants recruited via Prolific were recruited between the 5 and 10 May 2020. In these periods, all countries involved were under severe social restrictions (see below), and data collection was finished before countries started to lift lockdown restrictions. Out of the 1746 participants, 256 (104 from social media and 152 from Prolific) reported having been diagnosed in the past/present with a psychiatric

disorder and thus were removed from data analyses, resulting in a final sample size of 1490 participants (949 female, 539 male, 2 other; $M_{age} = 37.08$, s.d.$_{age} = 14.30$).

On average, people reported having been practising social distancing 46.41 days (s.d. = 10.56, range: 0–120 days; 10–90 percentile range = 35–60 days); with the level of regulations applied to their country being advice to not engage in social interactions (i.e. social distancing; $n = 71$); lockdown (advice to stay at home unless you need to go out, prohibition of social gatherings and interactions, fines in order for those who do not adhere to regulation; $n = 808$); and complete lockdown (e.g. prohibited to leave the house without a clear purpose, stay in a close radius of your house, most stores and businesses closed; $n = 611$). In terms of geographical location, 174 reported a country of residence within Latin America (e.g. Mexico, Argentina), 368 within continental Europe (e.g. France, Spain Greece), 892 in the UK and 56 outside the aforementioned regions (e.g. Australia, Canada, USA). With respect to household, 187 reported living alone, 438 reported living with one person, 306 with two, 241 with three, 193 with four, 86 with five and 39 with more than five people.

## 2.2. Procedure and measures

### 2.2.1. Procedure

After consenting to take part in the study, participants first completed questions about social isolation (note that when the study was carried out, all participants' countries of residence were in a certain degree of social distancing rules). This included the type of regulations in their country, to what degree participants were practising social distancing, days in lockdown and tolerance for isolation, followed by the short loneliness questionnaire [60] (see §2.2.4). Next, participants answered questions regarding their touch experience during COVID-19 (see §2.2.2), followed by the short version of the State–Trait Anxiety Inventory (STAI-SF; [61]), (see §2.2.3), questions about their experience and attitudes towards touch (see 2.2.6), and the self-report measure of adult attachment style, namely the ECR-S [55] (see §2.2.5). Finally, participants answered demographic questions (age, country of residence, nationality, psychiatric history and health). Measures of interest are detailed below, and all the others are described in the electronic supplementary material. Single items of mental health and tolerance for isolation were also collected and its relationship with touch deprivation was also examined (see electronic supplementary material). Correlations between variables of interest are summarized in electronic supplementary material, figure S1.

The default language of the online survey was English, but participants could choose among several languages (English, French, Spanish, Italian, Greek, Portuguese and Marathi). Each question was back-translated and for the main questionnaires, the validated translation in each language used (i.e. for STAI-SF and ECR-S). Note, that from the final sample of 1490 participants, 1130 participants took the questionnaire in English, 193 in French, 119 in Spanish, 24 in Italian and 24 in Greek.

### 2.2.2. Touch experience during COVID-19-related lockdown period

Using a visual analogue scale ranging from 0 'not at all' to 100 'a lot' for three items corresponding to different types of 'social' touch, (i.e. (i) intimate, e.g. kiss, hugs, caress from partner or close family; (ii) friendly, e.g. hugs and high-fives from friends or acquaintances; (iii) professional, e.g. handshakes, tap on the shoulder from colleagues, touch from carers), participants were asked to answer the following questions about their tactile experience: (i) 'Before COVID-19, How much touch of these different "social" types of touch were you getting?'; (ii) 'In the past week, How much touch of these different types of "social" touch have you been getting?'; (iii) 'In the past week, How much would you have wanted to experience these different types of "social" touch?'. (See figure 1 for an example) Note that the last two questions relate to the participant's amount of touch experienced and wanted, respectively, during a period of COVID-19-related restrictions. To obtain an index of lack of touch during this period, we subtracted the scores from question (ii) touch experienced in the past week from question (i) touch experienced before COVID-19, separately for each item (i.e. intimate, friendly, professional). See figure 2 for plots with means and error bars.

### 2.2.3. Anxiety

To examine anxiety, we used the well-validated six-item short-form of the state scale of the State–Trait Anxiety Inventory (STAI-SF; [61]). It comprises six items rated on a 4-point scale (1, not at all, and 4,

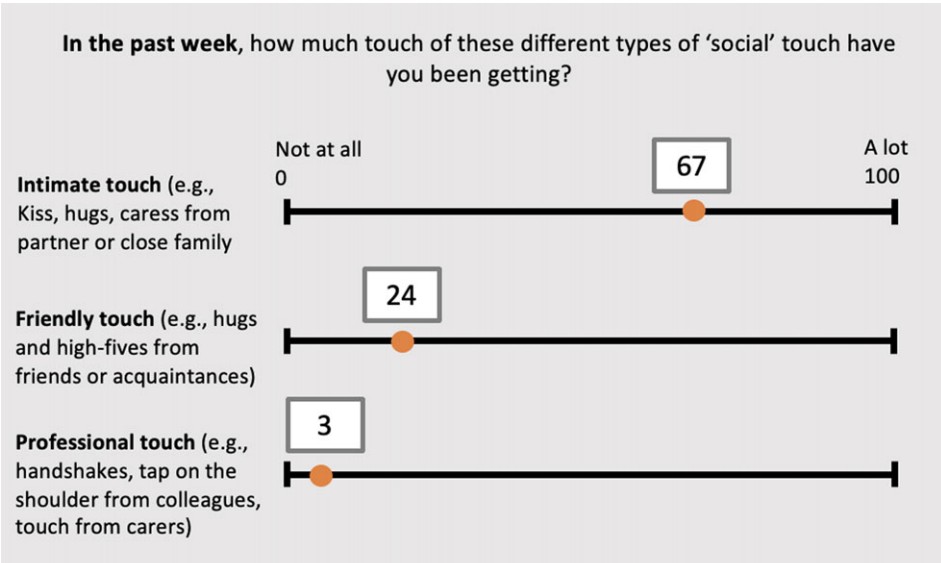

**Figure 1.** Example of the visual analogue scale ranging from 0 'not at all' to 100 'a lot' for the three items corresponding to different types of 'social' touch. The cursor was initially placed at 0, and participants had to move the cursor in order for the question to be validated. In this illustrative example, participants are asked about the amount of touch experienced in the past week (i.e. during COVID-19). However, participants were also asked about their tactile experience in relation to the amount of touch experienced before COVID-19 as well as their wanting to have experienced these types of touch in the past week.

very much). Items were summed (after reverse-scoring appropriate items) and multiplied by 20/6 to obtain an anxiety score. The STAI-SF is well validated and demonstrates very good internal consistency (Cronbach's $\alpha = 0.84$). The minimum and maximum possible score is 20 and 80, respectively. On average, the anxiety score was $M = 44.09$ (s.d. = 13.65). Higher scores denote more anxiety.

### 2.2.4. Loneliness

To examine feelings of loneliness, we used the validated, short, UCLA 3-item loneliness scale ('How often do you feel that you lack companionship?', 'How often do you feel left out?', 'How often do you feel isolated from others?') [60] as a measure of loneliness, as well as a single question asking 'How often do you feel lonely?', as recommended by the Office for National Statistics (2018). Items were rated on a 4-point (0, Never, and 3, Often) and a 5-point scale (1, Never, and 5, Often/Always), respectively. A summed score was computed for our measure of loneliness based on the three UCLA items, and demonstrated good internal consistency with the single question measure of loneliness, Cronbach's $\alpha = 0.79$. Thus, we averaged these loneliness scores to produce an index of loneliness for each participant, as advised by the NIH guidelines and the Office for National Statistics (2018), with higher scores denoting more perceived loneliness. Minimum and maximum possible score is 0.5 and 7, respectively. On average, loneliness score was $M = 3.61$ (s.d. = 1.62).

### 2.2.5. Self-report measure of adult attachment style (experiences in close relationships short—ECR-S)

As an index of adult attachment style, we collected the ECR-S [55] on the participants recruited from Prolific ($n = 1000$). The ECR-S comprises 12 items rated on a 7-point scale (1, strongly disagree, and 7, strongly agree) regarding the general experience of intimate adult relationships; 6 items pertain to attachment anxiety and 6 to attachment avoidance. Item responses are averaged (after reverse-scoring appropriate items) separately for each subscale to produce a mean score for attachment anxiety and attachment avoidance, with higher scores denoting greater attachment insecurity. The ECR-S is well validated [55] and demonstrates good internal consistency: Cronbach's $\alpha = 0.75$ for attachment anxiety and Cronbach's $\alpha = 0.80$ for attachment avoidance. The minimum and maximum possible score is 1 and 7, respectively. On average, the attachment anxiety score was $M = 3.72$ (s.d. = 1.18) and attachment avoidance $M = 2.67$ (s.d. = 1.09).

### 2.2.6. Attitudes and experiences towards touch

To examine attitudes and experiences towards touch, we used 7 items rated on a 5-point scale (1, disagree strongly, and 7, agree strongly) from the Touch Experiences and Attitudes Questionnaire (TEAQ; [57]); see electronic supplementary material for details on the selected items. Each item was selected as they correspond to one of the six components, and had the highest loading, from the TEAQ, namely friends and family touch (FFT), current intimate touch (CIT), childhood touch (ChT), attitude to self-care (ASC), attitude to intimate touch (AIT) and attitude to unfamiliar touch (AUT). Note that two items from the childhood touch component were included as they both corresponded to the highest loading in the original scale (i.e. 0.80). Moreover, we conducted a factor analysis (using the psych::fa function of R; [62]) on these items, and found that the two items from ChT do indeed correspond to the same factor (see electronic supplementary material, table S1). In addition, our factor analysis suggests that each item belongs to a separate factor or component, consistent with the original paper validating the TEAQ [57]. After reverse-scoring appropriate items, items demonstrated moderate internal consistency, with Cronbach's $\alpha = 0.63$. Items were summed to produce a total score for attitudes and experiences towards touch, with higher scores denoting more positive attitudes and experiences ($M = 24.31$, s.d. $= 5.27$). The minimum and maximum possible score is 7 and 49, respectively. Averaging across items, touch attitudes and experiences score was $M = 3.45$ (s.d. $= 0.76$).

## 2.3. Statistical analyses

All analyses were carried out in Stata 15. Given the big sample, we opted for a conservative approach of $p < 0.01$ rather than the conventional $p < 0.05$ to denote statistical significance. When sphericity was violated, the Greenhouse–Geisser correction was used where appropriate. Using the carr::vif R package, variance inflation scores (VIF) were calculated for each independent variable to make sure that there were in fact no multi-collinearity issues in our regression/mixed models. VIF scores above 5 indicate that there is a problematic amount of collinearity [63,64]. The following effect sizes for all analyses were computed using STATA: $\eta_P^2$ for repeated-measures ANOVA, $r$ for correlations and $\eta^2$ for regressions analyses. Marginal $R^2$ as well as conditional $R^2$ were computed for the multi-level regressions using the tab_model function of the R package sjPlot. The marginal $R^2$ considers only the variance of the fixed effects, while the conditional $R^2$ takes both the fixed and random effects into account.

### 2.3.1. Descriptive statistics and preliminary analyses

In order to characterize touch experience, we first conducted a repeated-measures ANOVA, specifying within-subjects factors of type of touch (intimate, friendly, professional) and time (before COVID-19 and in the past week, i.e. during COVID-19) on touch experience ratings. Next, we conducted a repeated-measures ANOVA, specifying within-subjects factors of type of touch (intimate, friendly, professional) on wanting touch (during COVID-19-related social restrictions). Interactions were followed up with $t$-tests where applicable. In particular, we expected the amount of touch experienced to be lower during COVID-19 relative to before and in particular friendly and professional touch, given social distancing restrictions.

Given COVID-19 restrictions, we also expected that the more the participants reported to practise social distancing, the more the lack of touch (for all types of touch but particularly friendly) as well as lack of touch to positively correlate with wanting touch, irrespective of the type of touch. This was examined with Pearson's correlations with the Bonferroni-adjusted $\alpha$-levels.

### 2.3.2. Main results

#### 2.3.2.1. Touch deprivation predicting loneliness and anxiety
To examine whether experiencing different types of social touch (i.e. professional, friendly and intimate) influences our wellbeing, we conducted multiple regressions separately on the following outcome variables: anxiety and loneliness. We entered computed scores of lack of touch in response to intimate, friendly and professional touch as continuous predictor variables (see §2.2.1) on how this difference score was computed. This difference score (rather than 'touch experienced in the past week') was used in order to account for differences in tactile experience before COVID-19-related restrictions. Although note that, as expected, we observe the same pattern of results, yet reversed, when looking at 'touch experienced in the past week' instead (see electronic supplementary material).

Continuous variables were mean-centred to avoid multi-collinearity issues and VIF scores were also checked (see above).

Given that anxiety and loneliness correlate (see electronic supplementary material, figure S1), we repeated the above regressions but this time also including the other outcome variable (anxiety or loneliness) as a predictor. This was done in order to check if our variables of interest explain a statistically significant account even after accounting for the other psychological index of wellbeing.

We expected the lack of intimate touch in particular to be associated with higher anxiety and feelings of loneliness.

### 2.3.2.2. Type of touch and days in lockdown predicting wanting touch

To examine whether wanting different types of touch in the past week was associated with days practising social distance, we specified a multi-level regression model with wanting touch ratings as the outcome variable and type of touch (intimate, friendly, professional), days practising social distancing, and their interaction, as predictor variables, and controlled for lack of touch by including it as a covariate. Days practising social distancing (continuous variable) was mean-centred in order to avoid multi-collinearity issues [65] and VIF scores were also checked (see above). A random effect was included to account for the repeated assessment of the outcome variable within individual.

We expected friendly touch in particular to be the most wanted during COVID-19 and to increase as a function of days practising social distancing.

### 2.3.2.3. Individual differences predicting wanting touch

First, to examine whether wanting different types of touch in the past week was associated with adult attachment style and attitudes and experiences towards touch, we specified a multi-level regression model with wanting touch ratings as the outcome variable and (i) attachment anxiety, attachment avoidance, type of touch (intimate, friendly, professional), and their interaction terms, as well as (ii) touch attitudes and experiences scores, type of touch (intimate, friendly, professional), and their interaction terms, as predictor variables, and controlled for lack of touch. Continuous variables were mean-centred in order to avoid multi-collinearity issues and VIF scores were also checked. A random effect was included to account for the repeated assessment of the outcome variable within individuals.

We expected that the more the attachment anxiety or attitudes and experiences towards touch, the more they would have wanted to experience touch in the past week; with the opposite pattern for attachment avoidance. To further examine whether wanting touch was associated with specific components of the attitudes and experiences towards touch measure, we specified another multi-level regression model with wanting touch ratings as the outcome variable and included the seven items of the experiences and attitudes towards touch measure as predictor variables.

# 3. Results

## 3.1. Descriptive statistics and preliminary analyses

### 3.1.1. Touch experience

As presented in figure 2, participants reported more touch experienced before COVID-19 ($M = 51.59$, s.d. $= 25.79$) when compared with the amount of touch reported in the past week ($M = 16.68$, s.d. $= 15.06$), $F_{1,1489} = 3306$, $p < 0.001$, $\eta_P^2 = 0.69$, irrespective of the type of touch. Intimate touch ($M = 52.18$, s.d. $= 32.67$) was reported as the most experienced, when compared with friendly ($M = 28.84$, s.d. $= 18.09$) and professional ($M = 21.38$, s.d. $= 17.51$) touch, $F_{1.4,2084.7} = 1018.2$, $p < 0.001$, $\eta_P^2 = 0.41$, irrespective of time. The type of touch interacted with time, $F_{1.763,2625.1} = 473.3$, $p < 0.001$, $\eta_P^2 = 0.24$. Touch was reported significantly less in the past week (versus before COVID-19) separately for intimate, friendly and professional touch, $ps < 0.001$. The type of touch by time interaction was driven by a larger difference between touch in the past week versus before COVID-19 in friendly ($M = 47.55$, s.d. $= 31.94$), when compared with intimate ($M = 19.85$, s.d. $= 32.10$) and professional touch ($M = 37.33$, s.d. $= 32.06$), $ps < 0.001$. Interestingly, we observe a similar pattern of results in response to wanting touch (during COVID-19-related social restrictions) as those reported to have experienced before COVID-19 (figure 2c). Specifically, the main effect of type of touch was statistically significant, $F_{1.894,2820.1} = 1281.2$, $p < 0.001$, $\eta_P^2 = 0.46$, with intimate touch ($M = 69.56$, s.d. $= 32.72$) being the most

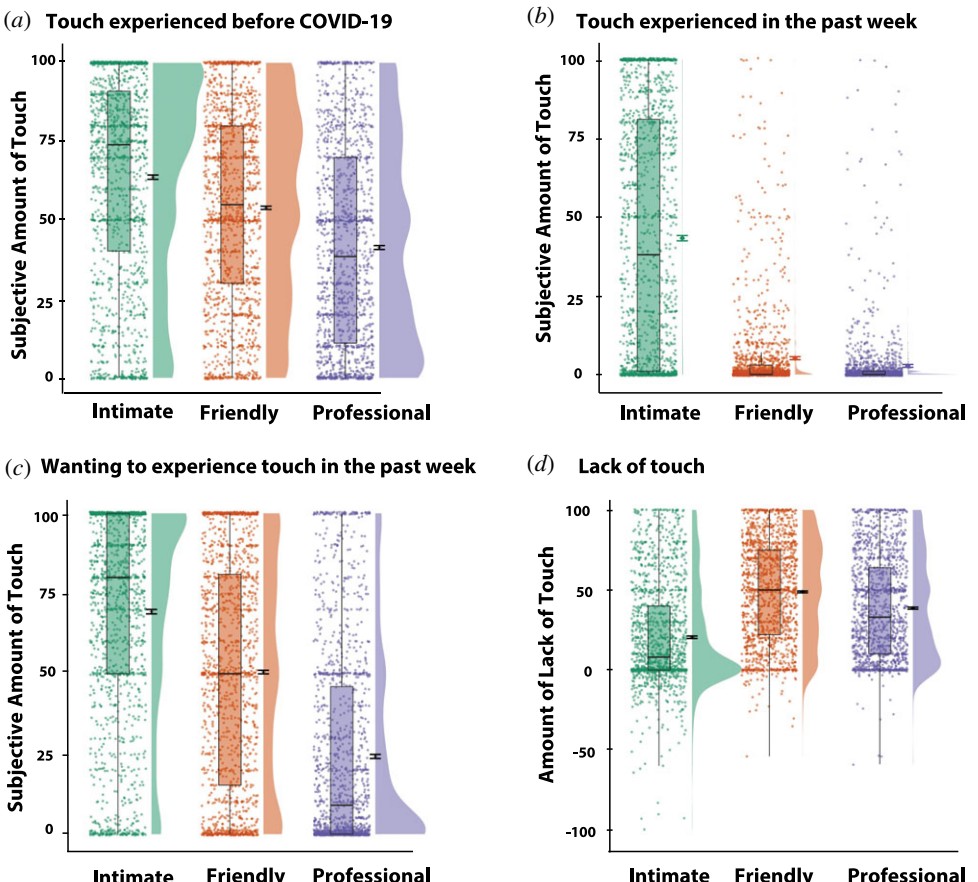

**Figure 2.** Ratings for touch experienced during COVID-19 for the three types of social touch: intimate, professional and friendly. (*a*) 'Before COVID-19, How much touch of these different types of 'social' touch were you getting?' from 0, not at all, to 100, a lot; (*b*) 'In the past week, How much touch of these different types of 'social' touch have you been getting?' from 0, not at all, to 100, a lot; (*c*) 'In the past week, How much would you have wanted to experience these different types of 'social' touch?' from 0, not at all, to 100, a lot. (*d*) Computed score for lack of touch during COVID-19: touch experienced in the last week was subtracted from touch experienced before COVID-19. Group distributions as unmirrored violin plots (probability density functions), individual data points, boxplot, mean and error bars denoting ±1 s.e.m.

wanted touch, when compared with friendly ($M = 50.58$, s.d. = 35.26) and professional ($M = 23.92$, s.d. = 30.61) touch ($ps < 0.001$). Note that the same pattern of effects remains when applying the Bonferroni correction in *post hoc* tests.

The number of household members or how close they feel with them does not influence the amount of touch people would have wanted to experience in the past week, i.e. during COVID-19. However, the closer they feel with the people they live with, the higher the amount of touch they report to have experienced, irrespective of before or during COVID-19-related restrictions. Interestingly, this is particularly the case for people living with more than five people before but not during COVID-19-related restrictions (see electronic supplementary material, tables S2 and S3 and figure S2).

### 3.1.2. Has practising social distancing resulted in a lack of touch?

There was a significantly weak positive correlation between practising social distancing and lack of touch for friendly, $r_{1490} = 0.09$, $p < 0.001$, but not intimate, $r_{1490} = 0.02$, $p = 0.515$, professional, $r_{1490} = 0.01$, $p = 0.663$, or their average, $r_{1490} = 0.05$, $p = 0.041$ (Bonferroni-corrected; figure 3*a*).

### 3.1.3. Does the lack of touch result in people wanting more touch?

There was a significant positive correlation between wanting touch in the past week and lack of touch, weak for intimate, $r_{1490} = 0.09$, $p < 0.001$, but strong for friendly, $r_{1490} = 0.48$, $p < 0.001$, professional, $r_{1490} = 0.46$, $p < 0.001$, and their average, $r_{1490} = 0.43$, $p < 0.001$ (Bonferroni-corrected; figure 3*b*).

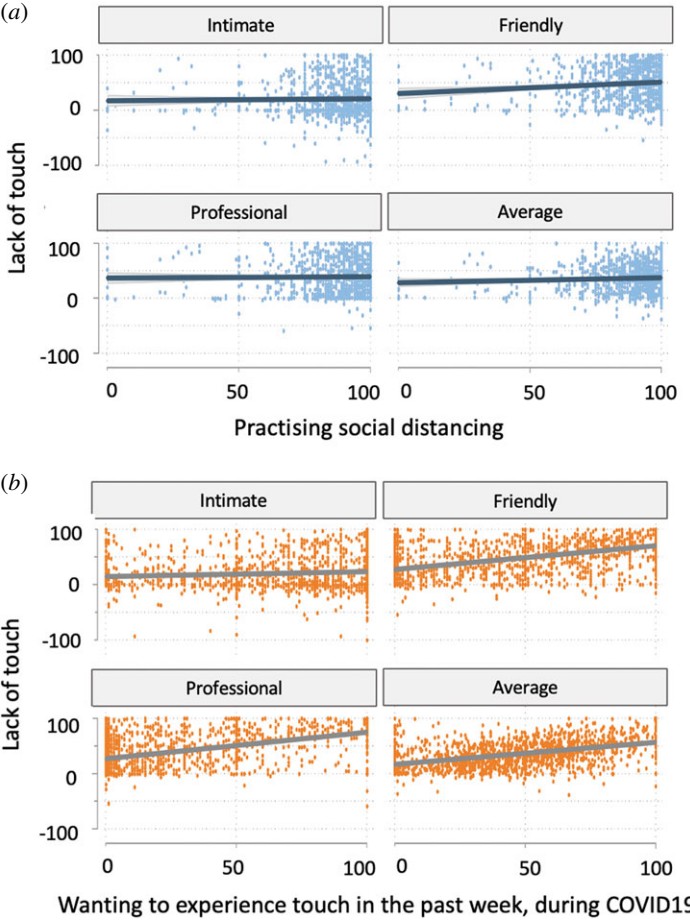

**Figure 3.** (*a*) Relationship between computed lack of touch (touch experienced before COVID-19 minus touch experienced in the past week) and the degree to which participants reported practising social distancing ('How much are you practicing social distancing?' ratings on a visual analogue scale ranging from 0 'not at all' to 100 'extremely') for intimate, friendly and professional touch, as well as their average. (*b*) Relationship between computed lack of touch and the amount of touch that participants would have wanted to experience in the past week for intimate, friendly and professional touch, as well as their average.

## 3.2. Main analyses

### 3.2.1. Does touch deprivation during COVID-19 influence feelings of loneliness and anxiety?

#### 3.2.1.1. Anxiety

A significant regression equation was found, $F_{3,1486} = 9.52$, $p < 0.001$, with an $R^2$ of 0.02. The lack of intimate touch was a significant predictor of anxiety, $b = 0.04$, s.e. = 0.01, $p < 0.001$, $\eta^2 = 0.009$, but not friendly, $b = 0.02$, s.e. = 0.01, $p = 0.083$, $\eta^2 = 0.002$, nor professional touch, $b = 0.01$, s.e. = 0.01, $p = 0.474$, $\eta^2 = 0.0003$ (figure 4). In sum, the more the lack of intimate touch, the more the anxiety (as expected, the same pattern of results yet in the opposite direction was observed when entering touch experienced in the past week instead; see electronic supplementary material, figure S3). Critically, when including loneliness in the model, the lack of intimate touch is no longer significant, $b = 0.01$, s.e. = 0.01, $p = 0.322$, $\eta^2 = 0.0007$ (friendly and professional, $p$s > 0.140). Note that a significant equation regression remains ($p < 0.001$) while $R^2$ is now of 0.21. VIF scores were all below 1.60.

#### 3.2.1.2. Loneliness

A significant regression equation was found, $F_{3,1486} = 14.49$, $p < 0.001$, with an $R^2$ of 0.03. The lack of intimate, $b = 0.01$, s.e. = 0.00, $p < 0.001$, $\eta^2 = 0.03$, but not friendly, $b = -0.00$, s.e. = 0.00, $p = 0.352$, $\eta^2 = 0.0006$, nor professional, $b = 0.00$, s.e. = 0.00 $p = 0.236$, $\eta^2 = 0.0009$, touch was a significant predictor of loneliness (figure 4). In sum, the more the lack of intimate touch experienced in the past week, the more

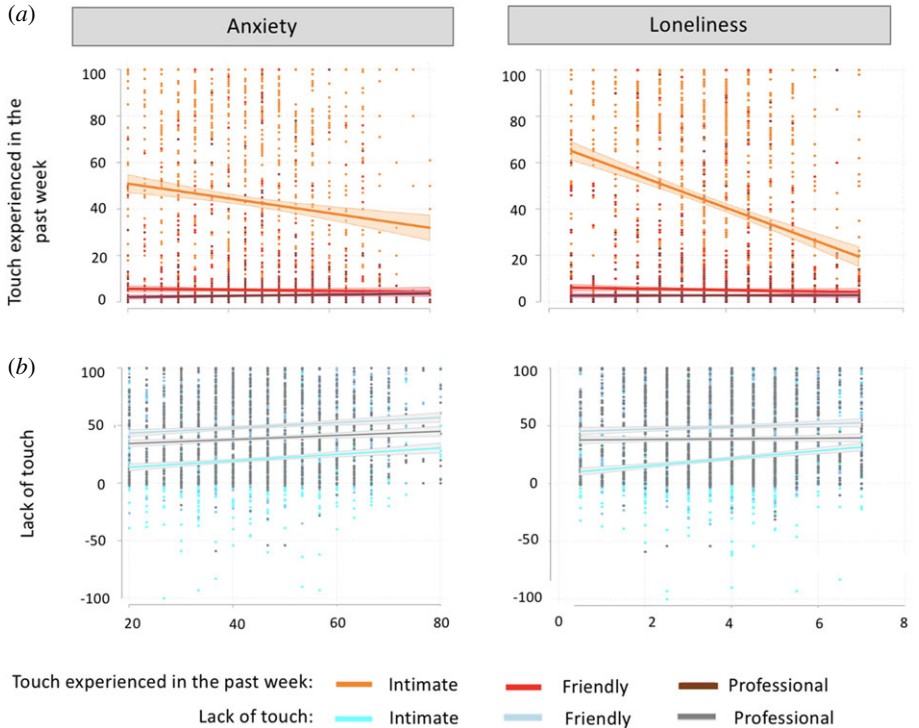

**Figure 4.** (a) Ratings in response to touch experienced in the past week (during COVID-19-related lockdown) in relation to anxiety and loneliness (see also electronic supplementary material). (b) Computed lack of touch (touch experienced before COVID-19 minus touch experienced in the past week) in relation to anxiety and loneliness.

the loneliness (the same pattern of results yet in the opposite direction, and larger $R^2$ of 0.09 was observed when instead entering touch experienced in the past week; see electronic supplementary material, figure S3). Critically, the same pattern of results were observed when including anxiety in the model, with the lack of intimate touch still predicting loneliness with the same strength, $b = 0.01$, s.e. $= 0.00$, $p < 0.001$, $\eta^2 = 0.02$ (friendly and professional, $p$s > 0.094). Note that a significant equation regression remains ($p < 0.001$), while $R^2$ is now of 0.22. VIF scores were all below 1.60.

### 3.2.2. What type of touch do people want to experience during COVID-19-related lockdown? And does this depend on days practising social distance?

The type of touch predicted wanting touch ratings: intimate touch ($M = 73.70$, s.d. $= 0.83$) was rated as more wanted than friendly ($M = 46.95$, s.d. $= 0.83$) and professional ($M = 23.47$, s.d. $= 0.81$) touch (see electronic supplementary material, table S4 for full model results). Days in lockdown also predicted wanting touch, $b = 0.20$, s.e. $= 0.08$, $p = 0.009$, in that the more the days in lockdown the more people want touch. Interestingly, the type of touch by days in lockdown interaction was also significant: friendly, $b = -0.24$, s.e. $= 0.08$, $p = 0.004$, and professional, $b = -0.34$, s.e. $= 0.08$, $p < 0.001$, touch by days in lockdown was significantly different to intimate touch (base level). Specifically, the more the days in lockdown, the more intimate touch people would want to experience in comparison with friendly and professional touch; figure 5. Note that the effect size for the full model is high: marginal $R^2 =$ 0.306 and conditional $R^2 = 0.586$. VIF scores were all below 1.66.

### 3.2.3. Does wanting (types of) touch during COVID-19 depend on individual differences?

As above, the type of touch predicted wanting touch ratings: intimate touch ($M = 70.67$, s.d. $= 1.00$) was rated as more wanted than friendly ($M = 43.58$, s.d. $= 1.00$) and professional touch ($M = 20.29$, s.d. $= 1.00$). Attachment anxiety, $b = 2.92$, s.e. $= 0.84$, $p < 0.001$, and attachment avoidance, $b = -10.86$, s.e. $= 1.03$, $p < 0.001$, predicted wanting touch: the more the attachment anxiety, the more the wanting to experience touch the past week; by contrast, the more the attachment avoidance the less the wanting to experience touch. Moreover, the type of touch by attachment anxiety interaction was significant: friendly, $b = 3.04$, s.e. $= 0.95$, $p < 0.001$, but not professional, $b = -0.97$, s.e. $= 0.95$, $p = 0.304$, touch by attachment anxiety was significantly different to intimate touch (base level). Specifically, the more the

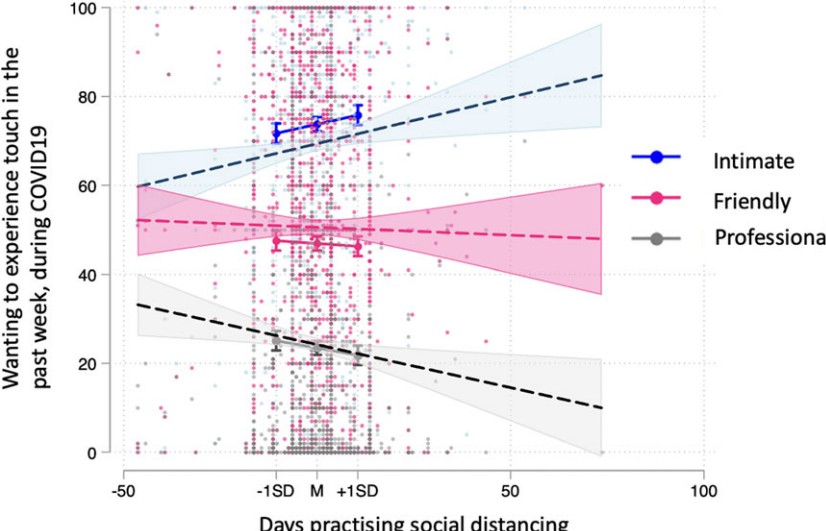

**Figure 5.** Interaction between type of touch and days practising social distancing. Follow-up continuous by categorical interactions were plotted using the 'margins' command at −1 s.d. (low), mean (moderate) and +1 s.d. (high) of continuous variables, in this case practising social distancing. Days practising social distancing were mean-centred.

attachment anxiety, the more friendly touch people want (figure 6a). The type of touch by attachment avoidance interaction was also significant: intimate touch by attachment avoidance was significantly different to friendly, $b = 10.04$, s.e. = 1.17, $p < 0.001$, or professional touch, $b = 11.25$, s.e. = 1.17, $p < 0.001$. Specifically, the more the attachment avoidance, the less intimate touch people want; figure 6b. In addition, there was a significant attachment avoidance by attachment anxiety interaction, $b = 1.99$, s.e. = 0.73, $p = 0.006$ (see electronic supplementary material, figure S4). The type of touch by attachment avoidance by attachment anxiety interaction was non-significant.

Attitudes and experiences towards touch predicted wanting touch, $b = 1.22$, s.e. = 0.21, $p < 0.001$, indicating that the more positive the touch experiences and attitudes, the more people would have wanted touch the past week. Importantly, the type of touch by touch experiences and attitudes interaction was significant: intimate touch by touch experiences and attitudes was significantly different to friendly, $b = 0.62$, s.e. = 0.23, $p = 0.006$, but not professional touch, $b = -0.18$, s.e. = 0.23, $p = 0.429$. Specifically, the more positive the touch attitudes and experiences, the more friendly touch people would want to experience; see figure 6c and electronic supplementary material, table S5 for full model results. Note that the effect size for the full model is large: marginal $R^2 = 0.429$ and conditional $R^2 = 0.635$. VIF scores were all below 1.91.

Interestingly, all the selected items from the TEAQ predicted wanting touch, $p$s < 0.001, except for the two items corresponding to the childhood touch experience (ChT) component, $p$s > 0.195, namely 'My parents were not very physically affectionate towards me during my childhood' and 'As a child my parents would tuck me up in bed every night and give me a hug and a kiss goodnight'. See electronic supplementary material, table S6 for full model results. Note that the effect size for the full model is medium: marginal $R^2 = 0.109$ and conditional $R^2 = 0.222$. VIF scores were all below 1.56.

## 4. Discussion

COVID-19-related restrictions inevitably affected core social habits of citizens, including tactile behaviours (with our data supporting this notion; see figures 2 and 3). Given growing laboratory and epidemiological evidence suggesting that social touch has beneficial effects on wellbeing [12,33,34], the present study first investigated whether the touch deprivation caused by COVID-19-related restrictions was associated with worse psychological outcomes. We found that the more the intimate touch (but not friendly or professional) experienced in the past week (i.e. during COVID-19), the better the targeted psychological outcomes: self-reported anxiety and feelings of loneliness (see electronic supplementary material for similar exploratory findings on single items measuring mental health and tolerating isolation), with the magnitude of these effects being small ($\eta^2 = 0.01$) and

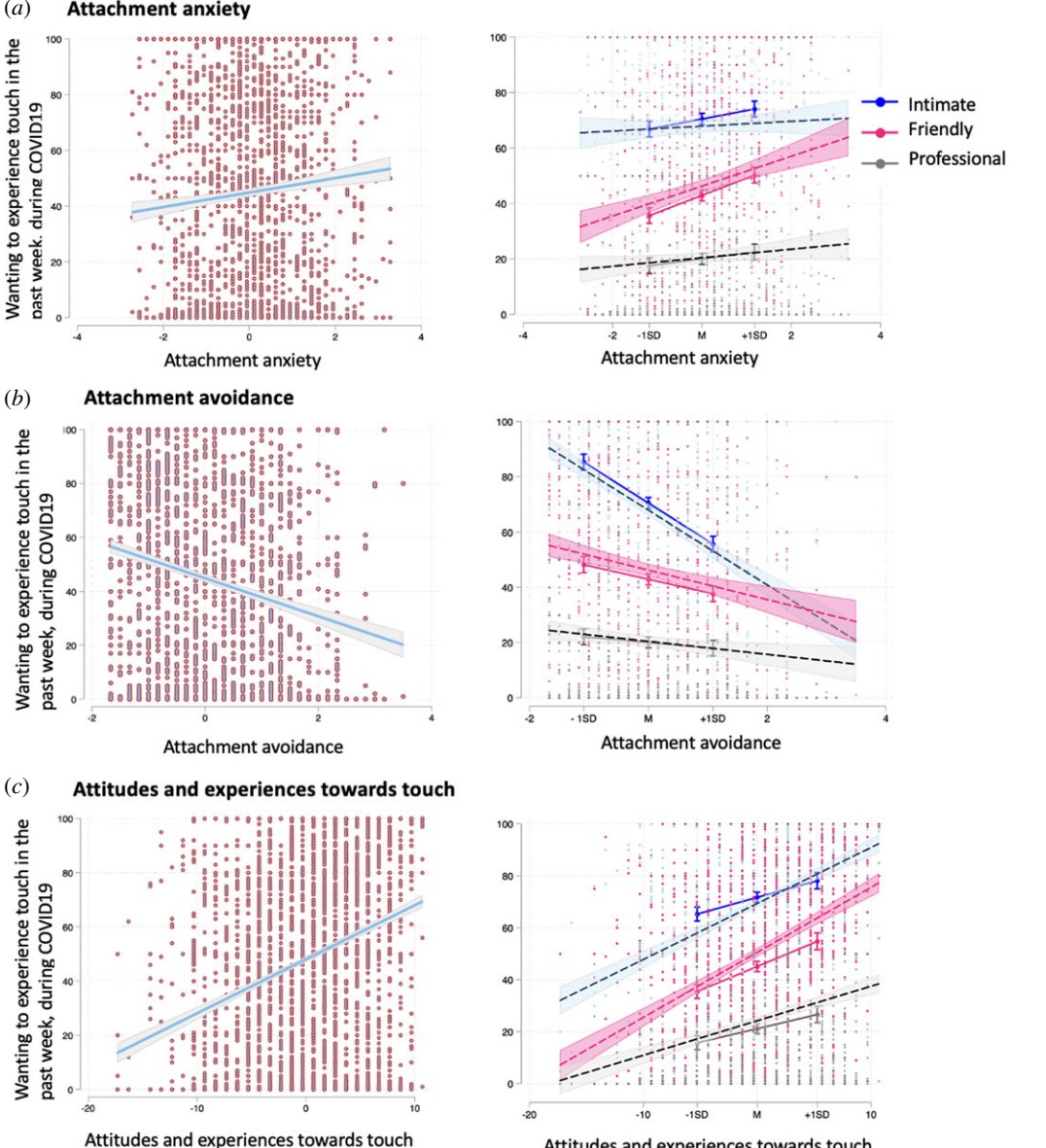

**Figure 6.** (*a*) Effects of attachment anxiety on wanting touch (left); interaction between types of touch and attachment anxiety (right). (*b*) Effects of attachment avoidance on wanting touch (left); interaction between types of touch and attachment avoidance (right). (*c*) Effects of attitudes and experiences towards touch on wanting to experience touch (left); interaction between types of touch and wanting to experience touch (right). Follow-up continuous by categorical (type of touch: intimate, friendly, professional) interactions were plotted using the 'margins' command at −1 s.d., mean and +1 s.d. of continuous variables.

moderate to large ($\eta^2 = 0.09$), respectively. These findings are consistent with growing evidence suggesting that the beneficial effects of touch are context-specific [6,39]. Indeed, touch is central to intimate, romantic relationships [66], and the regulatory role of touch seems to be mediated by psychological intimacy [51]. These findings are important, given that anxiety, depression and stress have been shown to be common reactions to the COVID-19 pandemic [50] and intimate touch may work as a protective factor. Interestingly though, as tested by this cross-section, self-report survey, while significant, the $R^2$ was low for anxiety (and mental health, tolerating isolation) although less so for loneliness (i.e. $R^2 = 0.09$; see electronic supplementary material), indicating that the latter is a better fit for the model yet only a small percentage of the variance can be explained by experienced touch. Given evidence suggesting that certain experiences that are likely to be experienced during COVID-19 may predict worse mental health outcomes (e.g. low income predicts mental distress [67] and illnesses or death of a close other predicts loneliness [68]), it is possible that other factors that determine anxiety and loneliness, that were not tested here (e.g. self-isolation history, conditions of work and income during lockdown), play a critical role in explaining variance.

Moreover, we found that the more the lack of intimate touch (but not friendly or professional), the worse the self-reported anxiety and feelings of loneliness. Importantly, unlike the above findings on touch experienced in the past week, the lack of touch computations take into account touch experienced before COVID-19 (i.e. baseline), thus making it specific to touch deprivation experienced during this period. For example, someone might be reporting little touch during COVID-19, but they might have been also experiencing little touch before, thus making it important to take these individual baselines into account. These findings are consistent with past research suggesting that when deprived of intimate touch, people show more mood and anxiety symptoms [48] and that those deprived of touch from close others report increased perceptions of loneliness [37]. Interestingly, we also found a moderate to strong correlation between anxiety and loneliness (electronic supplementary material, figure S1) and when accounting for the effect of loneliness on anxiety, our variable of interest (lack of intimate touch) no longer explains a statistically significant amount of variance. In other words, touch deprivation may not be a problem, at least for anxiety, unless one is lonely as well. Such finding has implications for touch research beyond the current pandemic effects. For example, future research on touch should consider loneliness when designing studies on touch effects. By contrast, when accounting for the effect of anxiety on loneliness, our variable of interest (lack of intimate touch) remained statistically significant. This suggests that the lack of touch during social distancing had effects on feelings of loneliness, even when controlling for related feelings of anxiety. Moreover, the latter model showed a higher $R^2$ (adjusted $R^2 = 0.22$) when including anxiety in the model versus not, indicating that the regression model fits the observed data better, explaining 22% of the variance. This is not surprising, given the tight relationship between touch and feelings of loneliness [10,37], but also between feelings of loneliness and anxiety (e.g. [69–71]). Taken together, these findings suggest that the effects of lack of intimate touch on loneliness go above and beyond the effects of touch on anxiety.

Interestingly, intimate (versus friendly and professional) touch was reported as the least deprived type of touch (figure 2); yet such effects seem to be the most pervasive on psychological wellbeing. Taken together, these findings suggest that experiencing intimate touch (e.g. kiss, hugs, caress from partner or close family) plays an important role on our wellbeing, and particularly on our feelings of loneliness, during COVID-19. Indeed, the effects of touch deprivation on loneliness have been shown to be particularly strong among single people, perhaps suggesting that lower loneliness among married people might be partly explained by the regular availability of physical contact [37]. Future studies should examine in finer detail whether these effects are driven by the precise nature of the touch relation, e.g. close family member versus partner, or by the type of touch itself, e.g. kisses versus hugs, or both (for example, by using the longing for interpersonal touch picture questionnaire [57]). Moreover, it is worthwhile noticing that only a weak positive correlation was found between the extent to which participants reported to practise social distancing and the lack of friendly (but not intimate or professional) touch. The fact that this correlation was weak could be at least partly explained by individuals living with friends and flat mates, from which they could have received friendly touch despite lockdown restrictions. Another possibility is that some people may have chosen, or were able because of circumstances (e.g. work colleagues that are also friends) to still meet and touch certain close friends despite practising social distancing more generally. Finally, some people may not habitually touch their friends and hence they may have not reported lack of friendly touch during social distancing.

Furthermore, we found that the more days practising COVID-19-related social distancing, the more individuals wanted to experience touch, with the magnitude of this effect being large (marginal $R^2 = 0.306$). Consistent with the literature suggesting that threats to social connection promote seeking of social reconnection and proximity [12,52,53] we hypothesized that the more people perceive threat to their social connections (which probably increases by time practising social distancing), the more it increases their craving/wanting of touch. In line with this thinking, however, we expected friendly touch in particular to increase as a function of days practising social distancing (as it is the type of touch that was found to lack the most during COVID-19; figure 2). Nevertheless, we found that the more the days practising social distancing, the more individuals wanted to experience intimate touch (note that we controlled for lack of touch in our analyses). One possible explanation is that as days practising social distancing increase, the less individuals seek or want friendly and professional touch as these sources of touch are less available to them (there was in fact a negative association between the amount of touch experienced in the past week and days practising social distancing, for friendly and professional touch only, $p$s < 0.001) and in turn crave intimate touch during the pandemic, as they have continued to experience it, relatively to a higher degree. Indeed, touch has context-specific rewarding properties, which are reinforced by experience [72].

Critically, we also found that the degree of wanting to experience touch in the past week (i.e. during COVID-19-related lockdown) further depended on individual differences in adult attachment style as well as attitudes and experiences towards touch, with the magnitude of this effect being large (marginal $R^2 = 0.429$). Specifically, the higher the attachment anxiety, the more individuals would have wanted to experience touch in the past week, and particularly, friendly touch. Given that anxious attachment is associated with craving closeness and reassurance from others [73], we hypothesize that particularly in a time when physical proximity with others is impaired, they would need and want more tactile support. The fact that anxious attachment is also characterized by worries about rejection and abandonment from close others [68] might in turn make them want more friendly rather than intimate touch. By contrast, we found that the higher the attachment avoidance, the less individuals would have wanted to experience touch in the past week, and particularly, intimate touch. Given that avoidant attachment is associated with a need for emotional distance and reduced proximity seeking [74,75], we hypothesize that they would want to experience less touch during COVID-19, particularly from intimate, close others. This finding is consistent with research suggesting that avoidant individuals prefer to cope with pain on their own. For example, a recent study suggests that embodied support from a romantic partner results in more pain [76], which may extend to touch. However, the available measures do not allow us to know if this pattern of wanting touch depending on attachment style is specific to COVID-19. It is indeed possible that this is a trait characteristic (i.e. present before COVID-19) and future studies should examine this issue. Indeed, past research has shown that attachment avoidance is associated with less frequent touch experience [15] and it is possible that such effects extend to wanting as well.

With respect to attitudes and experiences towards touch, we found that the more the positive attitudes and experiences towards touch, the more individuals would have wanted to experience touch in the past week (during COVID-19). This makes sense as the more positive people feel about touch, the more it might increase its wanting in times of deprivation. Notably, this was particularly the case for friendly touch, in line with the fact that this particular touch is associated with positive meaning [72] and the most deprived during this period. Surprisingly, we found that all items selected from the TEAQ [57] components predicted wanting touch in the past week, except for the childhood touch (ChT) items. This was unexpected because tactile exposure seems to relate to tactile, enjoyable experiences with close, familiar others [64,77], which may be related to attachment and early tactile experiences [56]. However, it is worthwhile noting that contrary to other measures (e.g. adult attachment interview), the Experiences in close relationships—revised (ECR-R) targets explicit adult romantic relationships, without focusing on childhood experiences with carers. As such, even though attachment representations are thought to remain relatively stable across the lifespan [78], it is possible that these may not relate, at least regarding tactile experiences (although see [55]).

Our findings should be considered in the light of their limitations and directions for future research.

First, the current study was based solely on self-report measures, and as such, there might be social desirability effects. For example, it is possible that participants reported more touch deprivation than they were actually experiencing, merely because they were primed, given the aim of the study. Relatedly, participants could have indicated a similar pattern for wanting touch in the past week to the amount of touch experienced before COVID-19 simply because it was prompted given the order of the questions. Future research should examine this matter by randomizing or counterbalancing these items.

Second, similar to other studies on touch, the amount of touch reported to have been experienced in the past week is probably influenced by the longing for touch (as can also be observed in figure 2), with such problem being particularly present in the current study as it relied on verbal labels for frequencies, which are more vague and ambiguous [79]. Specifically, our study relied on retrospective estimates of experience that may be influenced both by experience itself and one's own reflective biases, including the desire for touch. Even though we partly controlled for some of these biases by also measuring the amount of touch experienced before COVID-19 (subject also to reflective biases), comparisons between individuals becomes problematic as 'a lot' can mean largely different things between individuals. Moreover, the fact that the longing for touch may be intertwined with the frequency of response makes it difficult to distinguish the wish for touch from the amount of touch received in the responses provided. For example, people with a high wish for touch who have received a high amount of touch, but still would want more, will presumably report the same value as people who have a low wish for touch and have received very little touch. Thus, other measures of touch experience are needed to account for the possibility of these measures (estimates of experienced touch and one's desire for touch) not being independent. For example, other studies have relied on asking for the absolute frequency of touch (i.e. a guessed count [15,31,74]). As an absolute measure of touch

frequency (i.e. numerical), the values are comparable, although one may still question how accurately participants can remember each occasion of casual touch during an entire week. Moreover, some studies have, for example, used a combination of numerical and verbal frequencies [77], but without analysing the relationship between the two. Alternatively, studies could rely on ambulatory assessment methods that best capture ongoing experience and not retrospective reflections on one's experience [15].

Third, the TEAQ questionnaire [57] was adapted for this study due to time constraints and only items loading highly in previous factor analyses were administered. Thus, even though *post hoc* internal consistency and factor analyses confirmed the internal consistency and construct validity of the selected items, future studies should examine the criterion and content validity of this measure, as well as the reliability of our results. Similarly, single items (e.g. mental health, tolerance for isolation) were used to further explore, in supplementary analyses, the effects of touch deprivation on psychological wellbeing. Although these questions yield consistent results with our main analyses, their validity remains low and hence results should be treated with caution.

Fourth, future research should examine if the current effects depend on demographic variables, such as geographical location, age and gender. We did not examine this issue further as it was beyond the scope of this paper. However, we did examine whether these factors played a role on experiencing touch, with results suggesting that geographical location, age, gender or their interaction, do not play a role on experiencing touch during or before COVID-19 ($ps > 0.016$). This was surprising given past research suggesting that the diversity of interpersonal touch is higher in warmer, higher and less conservative countries [31].

Finally, touch experience during COVID-19 in clinical samples, such as patients diagnosed with anxiety or depression, remains to be fully examined. In our sample, out of the 256 participants excluded for reporting a diagnosed psychiatric condition: 54 reported anxiety, 55 depression, 41 reported both and 106 other psychiatric condition. While preliminary analyses indicate that there is no evidence to suggest that wanting touch during COVID-19 depends on these psychiatric categories, we found that participants reporting other psychiatric condition, as well as those reporting anxiety and depression, had the lowest scores in lacking touch during COVID-19. However, the main effect of anxiety and depression did not reach significance at our predetermined $p$-value of 0.01 (see electronic supplementary material, figure S5 and tables S7 and S8). Future research is still needed to examine this issue further in actual clinical samples, with the adequate tools to assess psychiatric conditions, before drawing any firm conclusions.

In sum, these findings corroborate and extend previous literature on the important role of touch, particularly by close, intimate others, in times of physical distancing, psychological distress and social pain. We show that intimate touch deprivation during COVID-19 is associated with worse psychological wellbeing, namely feelings of loneliness and anxiety. In addition to these effects on wellbeing, individuals seem to crave this type of intimate touch the most during COVID-19, with such effects being more prominent the more the days they have been practising social distancing. However, craving touch during COVID-19 depends on individual differences in attachment style as well as in attitudes and experiences towards touch.

Ethics. The study was approved by the Ethics Committee at University College London. All participants gave informed consent, and all collected data were anonymous.

Data accessibility. The data and analysis script that support the findings of this study are available on the Open Science Framework (https://osf.io/b46cs/).

Supplementary information and analyses are provided in electronic supplementary material [80].

Authors' contributions. M.v.M. and L.P.K. had the original idea for this study, and developed the study concept together with A.F. All authors contributed to the final study design. M.v.M. and L.P.K performed the data analysis and interpretation with additional input from A.F. M.v.M. drafted the manuscript, and all other authors provided critical revisions. All authors approved the final version of the manuscript for submission.

Competing interests. The authors report no competing interests.

Funding. This work was supported by a European Research Council Consolidator Award (ERC-2018-COG-818070) (to A.F.).

Acknowledgements. We would like to thank Alkistis Mavrogalou Foti and Ilia Galouzidi for the Greek translation, Elena Salvadé for the Italian translation, Paulo Silva for the Portuguese translation and Kohinoor Darda for the translation in Marathi. We thank also all the participants that took part and people who shared the survey helping us to gather all these data. We also would like to thank Athanasios Koukoutsakis for his help during the revision of this article.

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
