## [Peer Review File · Royal Society Open Science]

Review History

RSOS-210287.R0 (Original submission)

Review form: Reviewer 1

Is the manuscript scientifically sound in its present form?

No

Are the interpretations and conclusions justified by the results?

Yes

Is the language acceptable?

Yes

Do you have any ethical concerns with this paper?

No

Have you any concerns about statistical analyses in this paper?

Yes

Recommendation?

Major revision is needed (please make suggestions in comments)

Comments to the Author(s)

See attached PDF-file (Appendix A).

Review form: Reviewer 2**Is the manuscript scientifically sound in its present form?**

No

Are the interpretations and conclusions justified by the results?

Yes

Is the language acceptable?

Yes

Do you have any ethical concerns with this paper?

No

Have you any concerns about statistical analyses in this paper?

No

Recommendation?

Major revision is needed (please make suggestions in comments)

Comments to the Author(s)

This study investigated how the restrictions imposed to contain the covid-19 pandemic affected experienced touch, touch deprivation and its effects on psychological well-being. In an online survey, participants were asked about their touch experiences during the restrictions, mental health and craving for touch. Moreover, the authors assessed whether attachment style and attitudes towards touch affected their touch experiences. A relatively large sample size of 1746 individuals participated. The results showed that friendly touch was lacking most during the restrictions and that intimate touch was experienced most but was also most wanted during the restriction. Importantly, the longer the duration of the restrictions, the more individuals craved intimate touch. This was also dependent on attachment style, with opposite patterns for anxiously-attached and avoidant-attached styles. Moreover, lack of intimate touch was associated with higher levels of anxiety, loneliness, worse level of mental health and less tolerance of isolation. Finally, the more positive individuals attitudes were towards touch, the more they wanted to experience friendly and intimate touch.

This study provides interesting and novel information about the effects of the covid-19 restrictions on touch experiences and its effects on mental health. Strong points of the review are its sample size and the relatively specific questions about the experienced touch during and before the restrictions. I do have a number of questions with respect to the questions used, the analyses and the interpretation:

Methodology

- P. 7, 2.2.5: The way mental health was assessed seems very limited. Only 1 question was asked as a self-rating. Why not assess this with a more extensive standardized questionnaire? Also, while for loneliness and tolerance of isolation, the precise question is mentioned, this is not the case for mental health. Please add this question.

- Furthermore, to what extent do anxiety and mental health overlap?
- P. 7, 2.2.7 Attitudes and Experiences towards touch: Which items exactly were selected from each component and why these items?
- Methods, p. 6: Has type of relationship with members of household been recorded for those not living alone? Do number of household members and type of a relation play a role in lack of touch and wanting touch?
- Why use a 5-point scale for some questions (loneliness, mental health) while for others a VAS from 0-100 was used (tolerance to isolation)?
- Similarly, it is not entirely clear how practicing social distancing was measured. Did the participants rate this on a VAS from 1-100?

Results

Main analyses, 3.2.1, pp 10-11: the R2 for all regression analyses, while significant, are relatively low (.02 for all except loneliness). Thus only a small percentage of the variance can be explained by experienced touch. Perhaps this could be discussed in the Discussion

Minor comments

- P. 6, line 12: "368 within Europe" please change to "368 within continental Europe"
- Conclusion, p. 18: "social distancing" is mentioned, however according to the introduction this should be "physical distancing".
- Supplementary material contains a Figure S2 and a Table S6 which reported analyses concerning psychiatric conditions. These are only referred to in the Discussion of the manuscript, without mentioning the Figure and Table numbers. Please add those.

Decision letter (RSOS-210287.R0)

Dear Dr Kirsch

The Editors assigned to your paper RSOS-210287 "Social touch deprivation during COVID-19: effects on psychological wellbeing, tolerating isolation and craving interpersonal touch" have now received comments from reviewers and would like you to revise the paper in accordance with the reviewer comments and any comments from the Editors. Please note this decision does not guarantee eventual acceptance.

Please submit your revised manuscript and required files (see below) no later than 21 days from today's (ie 07-Apr-2021) date. Note: the ScholarOne system will 'lock' if submission of the

revision is attempted 21 or more days after the deadline. If you do not think you will be able to meet this deadline please contact the editorial office immediately.

on behalf of Dr Rochelle Ackerley (Associate Editor) and Essi Viding (Subject Editor)
openscience@royalsociety.org

Associate Editor Comments to Author (Dr Rochelle Ackerley):

Associate Editor: 1

Comments to the Author:

Articles on the effects of the COVID-19 pandemic are extremely important at the moment and the work presented here very much adds to this. The reviewers have raised a number of important points about the work, all of which need to be addressed to increase the clarity of the study and to justify the approach, especially the validity of the measures.

Reviewer comments to Author:

Reviewer: 1

Comments to the Author(s)

see attached PDF-file

Reviewer: 2

Comments to the Author(s)

This study investigated how the restrictions imposed to contain the covid-19 pandemic affected experienced touch, touch deprivation and it's effects on psychological well-being. In an online survey, participants were asked about their touch experiences during the restrictions, mental health and craving for touch. Moreover, the authors assessed whether attachment style and attitudes towards touch affected their touch experiences. A relatively large sample size of 1746 individuals participated. The results showed that friendly touch was lacking most during the restrictions and that intimate touch was experienced most but was also most wanted during the restriction. Importantly, the longer the duration of the restrictions, the more individuals craved intimate touch. This was also dependent on attachment style, with opposite patterns for anxiously-attached and avoidant-attached styles. Moreover, lack of intimate touch was associated with higher levels of anxiety, loneliness, worse level of mental health and less tolerance of isolation. Finally, the more positive individuals attitudes were towards touch, the more they wanted to experience friendly and intimate touch.

This study provides interesting and novel information about the effects of the covid-19 restrictions on touch experiences and it's effects on mental health. Strong points of the review are it's sample size and the relatively specific questions about the experienced touch during and

before the restrictions. I do have a number of questions with respect to the questions used, the analyses and the interpretation:

Methodology

- P. 7, 2.2.5: The way mental health was assessed seems very limited. Only 1 question was asked as a self-rating. Why not assess this with a more extensive standardized questionnaire? Also, while for loneliness and tolerance of isolation, the precise question is mentioned, this is not the case for mental health. Please add this question.
- Furthermore, to what extent do anxiety and mental health overlap?
- P. 7, 2.2.7 Attitudes and Experiences towards touch: Which items exactly were selected from each component and why these items?
- Methods, p. 6: Has type of relationship with members of household been recorded for those not living alone? Do number of household members and type of a relation play a role in lack of touch and wanting touch?
- Why use a 5-point scale for some questions (loneliness, mental health) while for others a VAS from 0-100 was used (tolerance to isolation)?
- Similarly, it is not entirely clear how practicing social distancing was measured. Did the participants rate this on a VAS from 1-100?

Results

Main analyses, 3.2.1, pp 10-11: the R2 for all regression analyses, while significant, are relatively low (.02 for all except loneliness). Thus only a small percentage of the variance can be explained by experienced touch. Perhaps this could be discussed in the Discussion

Minor comments

- P. 6, line 12: "368 within Europe" please change to "368 within continental Europe"
- Conclusion, p. 18: "social distancing" is mentioned, however according to the introduction this should be "physical distancing".
- Supplementary material contains a Figure S2 and a Table S6 which reported analyses concerning psychiatric conditions. These are only referred to in the Discussion of the manuscript, without mentioning the Figure and Table numbers. Please add those.

===PREPARING YOUR MANUSCRIPT===

===PREPARING YOUR REVISION IN SCHOLARONE===

Author's Response to Decision Letter for (RSOS-210287.R0)

See Appendix B.

RSOS-210287.R1 (Revision)

Review form: Reviewer 1

Is the manuscript scientifically sound in its present form?

Yes

Are the interpretations and conclusions justified by the results?

Yes

Is the language acceptable?

Yes

Do you have any ethical concerns with this paper?

No

Have you any concerns about statistical analyses in this paper?

No

Recommendation?

Accept with minor revision (please list in comments)

Comments to the Author(s)

See separate uploaded file (Appendix C).

Review form: Reviewer 2

Is the manuscript scientifically sound in its present form?

Yes

Are the interpretations and conclusions justified by the results?

Yes

Is the language acceptable?

Yes

Do you have any ethical concerns with this paper?

No

Have you any concerns about statistical analyses in this paper?

No

Recommendation?

Accept with minor revision (please list in comments)

Comments to the Author(s)

Overall, the authors have done a very good job revising the manuscript and have addressed my previous concerns. I have one remaining request: Figures 4, 5 and 6 are very difficult to read as the lines for the different types of touch are more or less obscured by the individual data points. It would be nice if this could be improved. Also for Figure 2, the scale of the y-axis is much smaller than that of the x-axis, which makes the significant relationships more difficult to detect.

Decision letter (RSOS-210287.R1)

Dear Dr Kirsch

On behalf of the Editors, we are pleased to inform you that your Manuscript RSOS-210287.R1 "Social touch deprivation during COVID-19: effects on psychological wellbeing and craving interpersonal touch" has been accepted for publication in Royal Society Open Science subject to minor revision in accordance with the referees' reports. Please find the referees' comments along with any feedback from the Editors below my signature.

Please submit your revised manuscript and required files (see below) no later than 7 days from today's (ie 02-Aug-2021) date. Note: the ScholarOne system will 'lock' if submission of the revision is attempted 7 or more days after the deadline. If you do not think you will be able to meet this deadline please contact the editorial office immediately.

on behalf of Dr Rochelle Ackerley (Associate Editor) and Essi Viding (Subject Editor)
openscience@royalsociety.org

Associate Editor Comments to Author (Dr Rochelle Ackerley):

I agree with the reviewers that the authors have done an excellent job of revising their manuscript and answering the comments. I am recommending that your paper is accepted, but some final minor revisions are required (more so from Reviewer 1). Reviewer 2 also states that the datafile is available and easy to understand, but they cannot find any data analyses syntax files, therefore could these please be made additionally available?

Reviewer comments to Author:

Reviewer: 1

Comments to the Author(s)

see separate uploaded file: "comments to revision.pdf".

Reviewer: 2

Comments to the Author(s)

Overall, the authors have done a very good job revising the manuscript and have addressed my previous concerns. I have one remaining request: Figures 4, 5 and 6 are very difficult to read as the lines for the different types of touch are more or less obscured by the individual data points. It would be nice if this could be improved. Also for Figure 2, the scale of the y-axis is much smaller than that of the x-axis, which makes the significant relationships more difficult to detect.

===PREPARING YOUR MANUSCRIPT===

===PREPARING YOUR REVISION IN SCHOLARONE===

<https://royalsociety.org/journals/authors/author-guidelines/#supplementary-material> to include a suitable title and informative caption. An example of appropriate titling and captioning may be found at https://figshare.com/articles/Table_S2_from_Is_there_a_trade-off_between_peak_performance_and_performance_breadth_across_temperatures_for_aerobic_scorpions_in_teleost_fishes_/3843624.

Author's Response to Decision Letter for (RSOS-210287.R1)

See Appendix D.

Decision letter (RSOS-210287.R2)

Dear Dr Kirsch,

I am pleased to inform you that your manuscript entitled "Social touch deprivation during COVID-19: effects on psychological wellbeing and craving interpersonal touch" is now accepted for publication in Royal Society Open Science.

COVID-19 rapid publication process:

We are taking steps to expedite the publication of research relevant to the pandemic. If you wish, you can opt to have your paper published as soon as it is ready, rather than waiting for it to be published the scheduled Wednesday.

This means your paper will not be included in the weekly media round-up which the Society sends to journalists ahead of publication. However, it will still appear in the COVID-19 Publishing Collection which journalists will be directed to each week (<https://royalsocietypublishing.org/topic/special-collections/novel-coronavirus-outbreak>).

If you wish to have your paper considered for immediate publication, or to discuss further, please notify openscience_proofs@royalsociety.org and press@royalsociety.org when you respond to this email.

on behalf of Dr Rochelle Ackerley (Associate Editor) and Essi Viding (Subject Editor)
openscience@royalsociety.org

Appendix A

This paper is on the subjective amount of received and desired touch and how these perceived amounts changed with social distancing during the Covid-19-pandemic. Changes in these variables are related to certain psycho-logical outcomes. The topic of this article is of interest to scholars in the field of touch and well-being, but also to the general public, and as such fits well into the scope of RSOS. However, the manuscript has in its present form some major problems that make it unsuitable for publication in RSOS. If the authors address the methodological shortcomings with a different analysis and increase the clarity of their manuscript, it may deliver timely insight into a question of general interest.

I outline a number of issues with the manuscript that need to be addressed by the authors.

1. Concerns about the validity of some of the measures used. The authors use several outcome measures that are not validated: single-item measures constructed for the purpose of this questionnaire (tolerance to isolation, mental health), or single items extracted from existing validated scales (UCLA loneliness scale, Touch Experiences and Attitudes Scale). Whereas the original scales where these measures were extracted from are validated, this does not apply to the single items. The validation was done with the entire set of questions. Extracting single items from a validated questionnaire and putting them into a new context is a substantial alteration, so the validation of the original instrument does no longer apply. Moreover, it becomes clear only from the acknowledgements that all these items have also been translated to different questionnaires. Thus, it is not clear what these items actually measure. For example, it seems likely that the “tolerance to isolation” item captures a very similar concept than the loneliness items. This has not only implications for the interpretation of the results, but also for issues with multicollinearity in the analyses. Figure S3 indeed shows that some of these measures are correlated. The authors write that they mean-centered the measures to deal with multicollinearity, but it is nowhere stated whether this removed multicollinearity or not.

To deal with this problem, the authors should perform a factor analysis of all the measures together, including the validated questionnaires that were used as a whole. They have the sample size to do this. This is expected to result in some key factors, for example, a loneliness factor or an attachment anxiety factor. Then all further analyses should be performed with these factors instead of with the single items (for example, mental health measured with a single item). This procedure would not only reduce the number of analyses, it would also slim down the mixed model analyses, and presumably also reduce the danger of multicollinearity. Last but not least, one would get a clearer picture of what it was that was measured in the first place.

2. Related to this, the validity also of the main measures of outcome is unclear. Instead of asking for the absolute frequency of touch interactions as was done in existing studies, the authors ask for subjective amounts, as the response format was “not at all – a lot”. Thus, for someone with a high need for touch even a large amount of hugs may not be sufficient. In that way, the questions on the amount of touch are coloured by the individual need/wanting for touch and the two are not independent from each other. This is also evident from the fact that they are correlated (at least for two of the touch types). I do not see much that the authors can do about this, but it needs to be discussed (as a limitation). Also, this point needs to be discussed in reference to other studies that asked for the absolute frequencies of touch (Bessler et al. 2020, *International Journal of Psychology*, Debrot et al. 2020, *Psychology Bulletin*, and possibly also Sorokowska et al. 2021, *Personality and Social Psychology Bulletin*). For example, participants in the Bessler et al. study report that they receive more hand shakes that they wish for, which fits nicely to the findings on friendly touch in the present study. Moreover, these existing studies and also others that deal with the effects touch deprivation (e.g. Sailer and Ackerley 2019) need to be addressed already in the introduction. Two of them are mentioned in the discussion, but not with regard to their results. So far the paper lacks an introduction into what is known about touch deprivation in adults.
3. In general, details are missing in many places of this manuscript. Remember that a manuscript should provide as many details on the methods and analyses so that other researchers can repeat the study. I will specify which information is missing further down.

4. Unless this is required by RSOS, which I think is not the case, I would recommend to restructure the manuscript. It would be much easier to read if the methods were not intermixed with the results. I would ask the authors to have all analyses in one place, all the results in one place and all the hypotheses in one place. This spares the reader from hopping back and forth in the manuscript in the search of the operationalisation or the hypothesis. In addition, the hypotheses are often missing. Please state explicitly where you had hypotheses, and where the analysis was exploratory.
5. The authors calculated three separate repeated-measures Anova with one factor (type of touch) each. However, this does not allow to capture interactions. For example, Figure 1 suggests that the subjective amount of friendly and professional touch decreased during the pandemic, but not the amount of intimate touch. This would be important to know and would become evident from an interaction. To visualise this, the authors should do an 2x3 Anova with the factors time point (pre-covid, during covid), and type of touch (intimate, friendly, professional). The variable “lack of touch” is redundant here, as it is a difference value that is less informative than the means of the original variables (a low value for lack of touch could indicate both a high amount of touch before and during covid, and low amount of touch before and during covid). As the need for touch is a different concept, this would still require a separate analysis, but one would halve the number of Anovas in any case. Possibly this could also be achieved with one single analysis with one nested factor (pre-covid/during covid nested in received touch). I would not know how to do that myself offhand, but encourage the authors to find out about it.

In the following, I will comment on further points in their order of appearance

Page 5, line 37. Please motivate your sample size.

Page 6, line -1. It would be interesting to not only read the mean number of days practicing social distancing, but also the range.

Page 6, line 19. Add information about how the participants were instructed (was there a question to introduce the VAS)? Also, were the numbers (0 to 100) visible? State also if the VAS-scale always started in the middle, at 50. This could explain why there is quite a high number of answers at 50 in Figure 1.

Page 6, line 44. Give the range of values possible for the STAI.

Measures, general. Name the order of administration of these different measures. The discussion of priming effects later on suggests that touch experiences were asked for first, but this is not stated explicitly anywhere. State that the questionnaires were translated and also say how this was done, e.g. if any guidelines were followed, if the questionnaires were back-translated, etc.

Page 7, first line. Name these three items.

Page 7, line 42. Name these seven items.

Page 8, first line “to examine the effects of childhood touch further”. Were there any explicit hypotheses? See point on hypotheses above.

Page 8, line -1. A mean of 24.31 seems impossible given that the scale was from 1 to 7. If this was a typo and the mean was at 2.4, this would speak for a rather negative attitude and would need to be discussed.

Page 8, paragraph ”other measures”. State for each measure how it was measured. The correlation results are results and should be removed from this paragraph which is on the methods.

Figure 1. “Amount of touch”. To be more precise, this measure should be called “subjective amount”. Increase the font size of the numbers. Adding the labels of the VAS-scale to the y-axis would further increase clarity. I would also appreciate additional box-plots to also illustrate the quartiles and outliers.

Page 9, line 38. Information about the correlations should be added to the methods. The correlations between lack of touch and degree of social distancing are rather low, which is surprising given the hypothesis and the large sample. This should be discussed.

Page 10, line 44. For all of these mixed models (which I suggest to recalculate based on the factor analysis), data needs to be provided that documents that there was no multicollinearity.

Page 11, first line. “We expected the same pattern of effects as above” – namely? Please collect all hypotheses at one place.

Results, general. To structure the results with headings that represent the research questions is very helpful.

The authors discuss the regression analyses with respect to slope and p-value, but never the explained variance or the effect size. The variance explained should also be discussed. For example, the variance in anxiety and loneliness explained by subjective touch amount is very low. For some of the measures, this could have to do with insufficient validity of the measure.

p-values are sometimes reported as $p < .000$, and sometimes as $p = .000$. The p-value is never exactly .000. The authors should stick to the guidelines of the journal for reporting p-values. (APA-style is $p < .001$).

Figure 3. Please increase the font size for the numbers on the axes.

Page 12, multilevel regression analysis. Please specify how lack of touch was controlled for and how the effect size (last line) was calculated.

Page 12, last paragraph, “touch by days in lockdown was significantly different”. Please provide the direction of the effect, also in other places.

Figure S3. The crossed out correlation coefficients are a bit hard to read. Maybe crossing them out in grey would work better, otherwise try to mark them differently.

Appendix B

RESPONSE TO REVIEWERS

Associate Editor: 1

Comments to the Author:

Articles on the effects of the COVID-19 pandemic are extremely important at the moment and the work presented here very much adds to this. The reviewers have raised a number of important points about the work, all of which need to be addressed to increase the clarity of the study and to justify the approach, especially the validity of the measures.

Response:

We thank the Associate Editor for their positive comment and giving us the chance to answer to the reviewers' points. We were pleased to read that the reviewers were positive about the manuscript, and we have now had the time to carefully evaluate and respond to the helpful comments provided by both reviewers. In our response to reviewers, we respond to the expert reviewers' critiques in a point-by-point manner and highlight the corresponding changes in the revised version of the manuscript (in blue).

Reviewer comments to Author:

Reviewer: 1

This paper is on the subjective amount of received and desired touch and how these perceived amounts changed with social distancing during the Covid-19-pandemic. Changes in these variables are related to certain psycho-logical outcomes. The topic of this article is of interest to scholars in the field of touch and well-being, but also to the general public, and as such fits well into the scope of RSOS. However, the manuscript has in its present form some major problems that make it unsuitable for publication in RSOS. If the authors address the methodological shortcomings with a different analysis and increase the clarity of their manuscript, it may deliver timely insight into a question of general interest. I outline a number of issues with the manuscript that need to be addressed by the authors.

1. Concerns about the validity of some of the measures used. The authors use several outcome measures that are not validated: single-item measures constructed for the purpose of this questionnaire (tolerance to isolation, mental health), or single items extracted from existing validated scales (UCLA Loneliness scale, Touch Experiences and Attitudes Scale). Whereas the original scales where these measures where extracted from are validated, this does not apply to the single items. The validation was done with the entire set of questions. Extracting single items from a validated questionnaire and putting them into a new context is a substantial alteration, so the validation of the original instrument does no longer apply. Moreover, it becomes clear only from the acknowledgements that all these items have also been translated to different questionnaires. Thus, it is not clear what these items actually measure. For example, it seems likely that the "tolerance to isolation" item captures a very similar concept than the loneliness items. This has not only implications for the interpretation of the results, but also for issues with multicollinearity in the analyses. Figure S3 indeed shows that some of these

measures are correlated. The authors write that they mean-centered the measures to deal with multicollinearity, but it is nowhere stated whether this removed multicollinearity or not. To deal with this problem, the authors should perform a factor analysis of all the measures together, including the validated questionnaires that were used as a whole. They have the sample size to do this. This is expected to result in some key factors, for example, a loneliness factor or an attachment anxiety factor. Then all further analyses should be performed with these factors instead of with the single items (for example, mental health measured with a single item). This procedure would not only reduce the number of analyses, it would also slim down the mixed model analyses, and presumably also reduce the danger of multicollinearity. Last but not least, one would get a clearer picture of what it was that was measured in the first place.

Response:

We thank the reviewer for these points that allowed us to clarify the validity of our measures, refocus our analysis and further test this validity in new factor and multicollinearity analyses.

First, given the reviewers suggestions regarding the possibility that single-items, randomly selected from bigger surveys, do not capture the domain of mental health concerns with enough granularity and precision, we now focus our main hypotheses on our longer measures of anxiety and loneliness, that were actually not selected randomly and have proven validity. Specifically, to measure anxiety we did use a validated short version of the STAI (6 items, Marteau & Becker, 1992) and to measure feelings of loneliness we used the validated, short, UCLA 3-item loneliness scale (Hughes, Waite, Hawkey & Cacioppo, 2004), which was specifically developed to measure loneliness in large surveys. In addition, we included an additional single question (i.e., “how often do you feel lonely”), as recommended by the NIH guidelines:

Measuring loneliness: guidance for use of the national indicators on surveys Methodological guidance on how to use the recommended loneliness questions for adults and children and how to interpret and report the findings (2018). Office for National Statistics.

Page 3: “Specifically, we recommend four questions to capture different aspects of loneliness. The first three questions are from the University of California, Los Angeles (UCLA) three-item loneliness scale. The wording of the UCLA questions and response options are taken from the English Longitudinal Study of Ageing. The last is a direct question about how often the respondent feels lonely, currently used on the Community Life Survey.”

Our analyses further confirmed that these measures of loneliness demonstrated a good internal consistency Cronbach’s $\alpha = 0.79$. Although please note that we get the exact same pattern of results when instead only using the UCLA 3-item loneliness scale. We have now provided more details about this measure in the manuscript as well as clarified that items were not selected randomly from existing questionnaires. We have now also

not drawn any main conclusions from analyses using single-items. Instead, these analyses appear only as exploratory in our SM.

Second, regarding the reviewer's concerns about multicollinearity issues, we have indeed mean-centred all our continuous variables to avoid multicollinearity issues (Tabachnick et al., 2007) and we have now tested specifically the multicollinearity of our predictors (i.e., lack of intimate touch, lack of friendly touch, lack of professional touch) in the main regressions on anxiety and loneliness. We computed the variance inflation factor (VIF) and all our variables were below 1.60. Note that the smallest possible value of VIF is 1 (absence of multicollinearity) and a VIF value that exceeds 5 indicates a problematic amount of collinearity (James et al., 2014). This has now been added throughout our manuscript and explained in the Methods section. Moreover, the VIF values were also calculated for all IVs in all mixed models (see point below), also indicating appropriately low levels of collinearity.

Third, given the correlation between anxiety and feelings of loneliness (Supplementary Figure S3, now Supplementary Figure S1), we have now controlled for each other in our analyses by including them as a predictor in hierarchical regressions (see section 2.3 B for more details). When accounting for loneliness on anxiety, our variable of interest (lack of intimate touch) no longer explains a statistically significant amount of variance. In contrast, our variable of interest on loneliness remained statistically significant even after accounting for anxiety. We have now included these analyses in the methods and results sections and these findings are now also addressed in the discussion (p. 17).

Finally, with respect to the analyses conducted on wanting touch during COVID-19 and individual differences, we used (a) the well-validated short version of the ECR (ECR-RS, 12-items) to measure attachment anxiety and avoidance dimensions, and (b) seven selected items from the validated touch experiences and attitudes questionnaire (TEAQ, Trotter et al., 2018). We acknowledge that we did not properly explain our selection strategy for this measure, and we do so now in the revised manuscript, as well as conducted a confirmatory factor analysis. Specifically, these items were not selected randomly, but rather based on their individual loading in the original validation study. One item with the highest loading was selected from each of the six components of the original TEAQ (and 2 items from the Childhood touch component, given that both items had the exact same highest loading). Nevertheless, to be sure that these items in fact correspond to separate factors or components when administered in this more brief form, we conducted a factor analysis using the `psych::fa` function of R (Revelle, 2021) in our data. As expected, the loadings of the two items from the childhood touch component (labeled below as `EAT_parents_coded` and `EAT_tuck_coded`) were found to be highest in the same factor (see Table below). In addition, the confirmatory PCA suggests that each item belongs to a separate component or factor, consistent with the original paper validating the TEAQ (Trotter et al., 2018). This factor analysis thus provides internal validation to the items selected from the TEAQ. We have included

details about the selected items and factor analysis in the methods section of the manuscript as well as in Supplementary Materials (Supplementary Methods 2, Table S1).

```
Factor Analysis using method = pa
Call: fa(r = dtTearq, nfactors = 6, rotate = "varimax", SMC = FALSE,
      fm = "pa")
standardized loadings (pattern matrix) based upon correlation matrix
```

	PA1	PA2	PA4	PA3	PA5	PA6	h2	u2	com
EAT_bath_CODED	0.01	1.00	0.04	0.00	0.06	0.05	1.00	0.00156	1.0
EAT_parents_coded	0.75	-0.01	0.03	0.09	0.08	0.08	0.59	0.41277	1.1
EAT_tuck_coded	0.73	0.03	0.08	0.04	0.09	0.12	0.57	0.43262	1.1
EAT_huggreet_coded	0.22	0.06	0.15	0.11	0.15	0.94	1.00	0.00205	1.3
EAT_skin_coded	0.10	0.04	0.97	0.10	0.10	0.14	1.00	0.00241	1.1
EAT_gettinghug_coded	0.17	0.06	0.10	0.03	0.97	0.14	1.00	0.00036	1.1
EAT_stranger_coded	0.11	0.00	0.10	0.98	0.03	0.10	1.00	0.00482	1.1

	PA1	PA2	PA4	PA3	PA5	PA6
SS loadings	1.20	1.00	1.00	0.99	0.99	0.96
Proportion Var	0.17	0.14	0.14	0.14	0.14	0.14
Cumulative Var	0.17	0.31	0.46	0.60	0.74	0.88
Proportion Explained	0.19	0.16	0.16	0.16	0.16	0.16
Cumulative Proportion	0.19	0.36	0.52	0.68	0.84	1.00

2. Related to this, the validity also of the main measures of outcome is unclear. Instead of asking for the absolute frequency of touch interactions as was done in existing studies, the authors ask for subjective amounts, as the response format was “not at all – a lot”. Thus, for someone with a high need for touch even a large amount of hugs may not be sufficient. In that way, the questions on the amount of touch are coloured by the individual need/wanting for touch and the two are not independent from each other. This is also evident from the fact that they are correlated (at least for two of the touch types). I do not see much that the authors can do about this, but it needs to be discussed (as a limitation). Also, this point needs to be discussed in reference to other studies that asked for the absolute frequencies of touch (Bessler et al. 2020, International Journal of Psychology, Debrot et al. 2020, Psychology Bulletin, and possibly also Sorokowska et al. 2021, Personality and Social Psychology Bulletin). For example, participants in the Bessler et al. study report that they receive more hand shakes that they wish for, which fits nicely to the findings on friendly touch in the present study. Moreover, these existing studies and also others that deal with the effects touch deprivation (e.g. Sailer and Ackerley 2019) need to be addressed already in the introduction. Two of them are mentioned in the discussion, but not with regard to their results. So far the paper lacks an introduction into what is known about touch deprivation in adults.

Response:

Thank you. We had indeed, regrettably, neglected to mention this relevant literature and we have now added more reference to the existing literature on the effects of touch deprivation in the introduction as follows:

P. 4:

“As such, it is not surprising that touch deprivation is associated with negative outcomes. For example, in children, touch deprivation is associated with struggles in learning to speak (45), sleep problems and school performance (46) and aggression (47). In adults, touch deprivation is associated with higher mood and anxiety symptoms (48) depression (49) perceived loneliness (37) and worse wellbeing more generally (15).”

Our findings are now also discussed in relation to the literature highlighted by the reviewer throughout the discussion. Moreover, the mentioned limitation of asking for subjective amounts of touch is now discussed and addressed as a limitation in the discussion. Note that while we state our measures of touch as retrospective and subject to reflective biases, we think that both measures, ours and those using absolute frequencies of touch, are subjective. We now explain their differences and the advantages and limitations of our method as follows:

p. 19:

“Second, similarly to other studies on touch, the amount of touch reported to have been experienced in the past week is likely influenced by the longing for touch (as can also be observed in Figure 2) and vice versa, indicating a strong relationship between the two. Supportive of this notion, a recent study has shown that individuals report receiving more handshakes than they wish for (79), which fits nicely with our findings on friendly and professional touch (see Figure 1A and 1C). Thus, to account for the possibility of these measures (estimates of experienced touch and one’s desire for touch) not being independent, other studies have relied on asking for the absolute frequency of touch (i.e., a guessed count (15,31,74)). In contrast, our study relied on retrospective estimates of experience that may be influenced both by experience itself and one’s own reflective biases, including the desire for touch. However, by also measuring the amount of touch experienced before COVID-19 (subject also to reflective biases), we can at least partly control for some of these biases. To this end, our main analyses relied on a differential between how much touch was experienced before and during COVID-19 (i.e. lack of touch). Future studies could rely on ambulatory assessment methods that best capture ongoing experience and not retrospective reflections on one’s experience.”

3. In general, details are missing in many places of this manuscript. Remember that a manuscript should provide as many details on the methods and analyses so that other researchers can repeat the study. I will specify which information is missing further down.

Response: Along the reviewer’s comments below, we now have added more details all along the manuscript.

4. Unless this is required by RSOS, which I think is not the case, I would recommend to restructure the manuscript. It would be much easier to read if the methods were not intermixed with the results. I would ask the authors to have all analyses in one place, all the results in one

place and all the hypotheses in one place. This spares the reader from hopping back and forth in the manuscript in the search of the operationalisation or the hypothesis. In addition, the hypotheses are often missing. Please state explicitly where you had hypotheses, and where the analysis was exploratory.

Response: We thank the reviewer for this point. We have changed the structure of the manuscript and now have put everything in one place. We have also now stated more clearly the hypotheses, which can be found in the introduction, but are also now included in the analyses' plan in relation to each analysis in the Methods section (see p. 8 - 9).

5. The authors calculated three separate repeated-measures Anova with one factor (type of touch) each. However, this does not allow to capture interactions. For example, Figure 1 suggests that the subjective amount of friendly and professional touch decreased during the pandemic, but not the amount of intimate touch. This would be important to know and would become evident from an interaction. To visualise this, the authors should do an 2x3 Anova with the factors time point (precovid, during covid), and type of touch (intimate, friendly, professional). The variable "lack of touch" is redundant here, as it is a difference value that is less informative than the means of the original variables (a low value for lack of touch could indicate both a high amount of touch before and during covid, and low amount of touch before and during covid). As the need for touch is a different concept, this would still require a separate analysis, but one would halve the number of Anovas in any case.

Possibly this could also be achieved with one single analysis with one nested factor (pre-covid/during covid nested in received touch). I would not know how to do that myself offhand, but encourage the authors to find out about it.

Response: We thank the reviewer for their comment. We have changed these analyses as suggested. That section now reads as follows, in methods and results, respectively:

p. 9:

A. Descriptive statistics and preliminary analyses

"In order to characterise touch experience we first conducted a repeated measures ANOVA, specifying within-subjects factors of type of touch (intimate, friendly, professional) and time (before COVID-19 and in the past week, i.e., during COVID-19) on touch experience ratings. Next, we conducted a repeated measures ANOVA, specifying within-subjects factors of type of touch (intimate, friendly, professional) on wanting touch (during COVID-19-related social restrictions). In particular, we expected the amount of touch experienced to be lower during COVID-19 relative to before and in particular friendly and professional touch given social distancing restrictions."

p. 10:

"As presented in Figure 1, averaging across all types of touch, participants reported more touch experienced before COVID19 ($M=51.59$, $SD=25.79$) as compared to the amount of touch reported in the past week ($M=16.68$, $SD =15.06$), $F(1, 1489) = 3306$, $p < .001$, $\eta_p^2 = .69$. Across time, intimate touch ($M=52.18$, $SD=32.67$) was reported as the most experienced, as compared to

friendly ($M=28.84$, $SD=18.09$) and professional ($M=21.38$, $SD=17.51$) touch, $F(1.4, 2084.7) = 1018.2$, $p < .001$, $\eta_p^2 = .41$. The type of touch interacted with time, $F(1.763, 2625.1) = 473.3$, $p < .001$, $\eta_p^2 = .24$. The type of touch by time interaction was driven by a larger difference in friendly touch reported in the past week vs. before COVID-19 ($M=47.55$, $SD=31.94$), as compared to intimate ($M=19.85$, $SD=32.10$) and professional touch ($M=37.33$, $SD=32.06$), p 's $< .001$. Interestingly, we observe a similar pattern of results in response to wanting touch (during COVID-19-related social restrictions) as those reported to have experienced before COVID-19 (see Figure 1C). Specifically, the main effect of type of touch was statistically significant, $F(1.894, 2820.1) = 1281.2$, $p < .001$, $\eta_p^2 = .46$, with intimate touch ($M=69.56$, $SD=32.72$) being the most wanted touch, as compared to friendly ($M=50.58$, $SD=35.26$) and professional ($M=23.92$, $SD=30.61$) touch (p 's $< .001$). Note that the same pattern of effects remains when applying Bonferroni correction."

In the following, I will comment on further points in their order of appearance

Page 5, line 37. Please motivate your sample size.

Response: We have now motivated our sample size in the Methods section as follow: "This was a sample size of convenience based on a survey distributed as widely as possible within a given period of COVID-19 restrictions."

Page 6, line -1. It would be interesting to not only read the mean number of days practicing social distancing, but also the range.

Response: We have now added this information in the manuscript: "range 0 - 120 days" and "(10-90 percentile range = 35-60)".

Note that only <10 people answered 0, and less than 10% said more than 60 days, suggesting that the latter participants choose to self-isolate before governmental restrictions, please see also histogram and table with percentiles below. Please also note that while there are some outliers, i.e., 36 using a +/-3 SD criteria, we obtain the same pattern of results throughout when excluding these outliers from analyses. Thus, we took a conservative approach, and no outliers were excluded in the main analyses.

days_lockdown

```

      type:  numeric (int)

      range:  [0,120]                units:  1
unique values: 64                  missing .: 2/1,490

      mean:   46.4147
      std. dev: 10.5631

percentiles:    10%    25%    50%    75%    90%
                35     40     46     50     60

```

Page 6, line 19. Add information about how the participants were instructed (was there a question to introduce the VAS)? Also, were the numbers (0 to 100) visible? State also if the VAS-scale always started in the middle, at 50. This could explain why there is quite a high number of answers at 50 in Figure 1.

Response: We have now added more details about this. There was no preamble on how to use the VAS scale, but the question and presentation was explicit as you can see on an example below; showing the anchor of the scale (0=not at all to 100=A lot). The cursor was initially placed at 0, and participants had to move the cursor in order for the question to be validated. Thus, it cannot explain the fact that many participants put the cursor at 50 in Figure 1.

Page 6, line 44. Give the range of values possible for the STAI.

Response: The STAI is on a 4-point scale (Not at all, Somewhat, moderately so, very much so). We have now added these details in the manuscript.

Measures, general. Name the order of administration of these different measures. The discussion of priming effects later on suggests that touch experiences were asked for first, but this is not stated explicitly anywhere. State that the questionnaires were translated and also say how this was done, e.g. if any guidelines were followed, if the questionnaires were back-translated, etc.

Response: We have now added a separate paragraph in the method section explaining the procedure (including order of measures) and we added all this information in detail in the Supplementary Materials (including the OSF link to the full questionnaire exactly as it was delivered, <https://osf.io/b46cs/>).

Page 7, first line. Name these three items.

Response: We have now added these three items in the Methods Section (2.2.4, p.7).

Page 7, line 42. Name these seven items.

Response: We have now named these seven items, as well as the factor analysis conducted on these, in the Supplementary Materials.

Page 8, first line "to examine the effects of childhood touch further". Were there any explicit hypotheses? See point on hypotheses above.

Response: Thank you, we have now changed this and moved the hypotheses in one place as suggested (in the methods section).

Page 8, line -1. A mean of 24.31 seems impossible given that the scale was from 1 to 7. If this was a typo and the mean was at 2.4, this would speak for a rather negative attitude and would need to be discussed.

Response: We apologize for the confusion.

“Items were summed to produce a mean score for attitudes and experiences towards touch, with higher scores denoting more positive attitudes and experiences. On average, touch attitudes and experiences score was $M=24.31$ ($SD=5.27$).” **was supposed to read as follows:** *“Items were summed to produce a **total** score for attitudes and experiences towards touch, with higher scores denoting more positive attitudes and experiences ($M=24.31$, $SD=5.27$). **Averaging across items**, touch attitudes and experiences score was $M=3.45$ ($SD=.76$).”* **This has now been amended in the manuscript. These values are in line with previous work (e.g., Trotter, McGlone, Reniers, & Deakin, 2018).**

Page 8, paragraph “other measures”. State for each measure how it was measured. The correlation results are results and should be removed from this paragraph which is on the methods.

Response: Following the above comments, we have now added all details in the new part (Procedure), and full details are now included in the Supplementary Materials.

Figure 1. “Amount of touch”. To be more precise, this measure should be called “subjective amount”. Increase the font size of the numbers. Adding the labels of the VAS-scale to the y-axis would further increase clarity. I would also appreciate additional box-plots to also illustrate the quartiles and outliers.

Response: We have now redone the figure, to increase font size, and added box-plots for further illustration of the data. To not overload the figure, we added in the legend the fact that this represents a subjective amount of touch ratings, and added the anchor of the scale (from 0=not at all, to 100=a lot).

Figure 1. *Subjective* ratings for touch experienced during COVID-19 for the three types of social touch: intimate, professional and friendly. (A) “Before COVID-19, How much touch of these different ‘social’ types of touch were you getting?” from 0=not at all, to 100=a lot; (B) “In the past week, How much touch of these different types of ‘social’ touch have you been getting?” from 0=not at all, to 100=a lot; (C) “In the past week, How much would you have wanted to experience these different types of ‘social’ touch?” from 0=not at all, to 100=a lot. (D) Computed score for lack of touch during COVID-19: touch experienced in the last week was subtracted from touch experienced before COVID-19. Group distributions as un-mirrored violin plots (probability density functions), individual data points, boxplots, mean and error bars denoting ± 1 SEM.

Page 9, line 38. Information about the correlations should be added to the methods. The correlations between lack of touch and degree of social distancing are rather low, which is surprising given the hypothesis and the large sample. This should be discussed.

Response: Thank you. We have now added the information about the correlations in the methods section (see 2.3 Statistical analyses).

“Given COVID-19 restrictions, we also expected that the more the practicing social distancing, the more the lack of touch (for all types of touch but particularly friendly) as well as lack of touch to positively correlate with wanting touch, irrespective of the type of touch. This was examined with Pearson’s correlations with Bonferroni adjusted alpha levels.”

As suggested, we have discussed this weak correlation between practicing social distancing and friendly touch as follows in the discussion section of the manuscript:

p. 18:

“Moreover, it is worthwhile noticing that only a weak positive correlation was found between the extent to which participants reported to practice social distancing and the lack of friendly (but not intimate or professional) touch. The fact that this correlation was weak could be at least partly explained by individuals living with friends and flat mates, from which they could have received friendly touch despite lockdown restrictions. Another possibility is that some people may have chosen, or were able because of circumstances (e.g. work colleagues that are also friends) to still meet and touch certain close friends despite practicing social distancing more generally. Finally, some people may not habitually touch their friends and hence they may have not reported lack of friendly touch during social distancing.”

Page 10, line 44. For all of these mixed models (which I suggest to recalculate based on the factor analysis), data needs to be provided that documents that there was no multicollinearity.

Response: We have now provided data that demonstrates there was no multicollinearity in any of our models. This was done by calculating variance inflation scores (VIF) for each independent variable to make sure there were in fact no multicollinearity issues. As included in the statistical analyses section, a VIF of 5 or more indicates a problematic amount of collinearity (James et al., 2014). In all our models, no IV showed a VIF score higher than 1.91, indicating there were no multicollinearity issues.

Page 11, first line. “We expected the same pattern of effects as above” – namely? Please collect all hypotheses at one place.

Response: Thank you. We have now collected all hypotheses at one place.

Results, general. To structure the results with headings that represent the research questions is very helpful.

Response: Thank you.

The authors discuss the regression analyses with respect to slope and p-value, but never the explained variance or the effect size. The variance explained should also be discussed. For example, the variance in anxiety and loneliness explained by subjective touch amount is very low. For some of the measures, this could have to do with insufficient validity of the measure.

Response: We have now added more details in the methods and results section about the effect sizes and their small, medium or large levels. Effect sizes, which were all found to be medium to large or large are also now included in the discussion.

Moreover, we have now addressed the low variance explained in our regression models, as well as an observed higher R^2 when we include anxiety in our regression model on loneliness (and vice versa). See also comment from reviewer 2. It now reads as follows:

P 16-17: “ COVID-19-related restrictions inevitably affected core social habits of citizens, including tactile behaviours (with our data supporting this notion; see Figure 1 and Figure 2). Given growing lab and epidemiological evidence suggesting that social touch has beneficial effects on well-being (12,33,34), the present study first investigated whether the touch deprivation caused by COVID-19-related restrictions was associated with worse psychological outcomes. We found that the more the intimate touch (but not friendly or professional) experienced in the past week (i.e., during COVID-19), the better the targeted psychological outcomes: self-reported anxiety and feelings of loneliness (see Supplementary Material for similar exploratory findings on single items measuring mental health and tolerating isolation), with the magnitude of this effect being small ($\eta^2=.01$) and moderate to large ($\eta^2=.09$), respectively. These findings are consistent with growing evidence suggesting that the beneficial effects of touch are context-specific (6,39). Indeed, touch is central to intimate, romantic relationships (66), and the regulatory role of touch seems to be mediated by psychological intimacy (51). These findings are important given that anxiety, depression and stress have been shown to be common reactions to the COVID-19 pandemic (50) and intimate touch may work as a protective factor. Interestingly though, while significant, the R^2 was low for anxiety (and mental health, tolerating isolation) although less so for loneliness (i.e., $R^2 =.09$; see Supplementary materials), indicating that the latter is a better fit for the model yet only a small percentage of the variance can be explained by experienced touch. Given evidence suggesting that certain experiences that are likely to be experienced during COVID-19 may predict worse mental health outcomes (e.g., low income predicts mental distress (67) and illnesses or death of a close other predicts loneliness (68)), it is possible that other factors that determine anxiety and loneliness, that were not tested here (e.g., self-isolation history, conditions of work and income during lockdown) play a critical role in explaining variance.

Moreover, we found that the more the lack of intimate touch (but not friendly or professional), the worse the self-reported anxiety and feelings of loneliness. Importantly, unlike the above findings on touch experienced in the past week, lack of touch computations take into account touch experienced before COVID-19 (i.e., baseline), thus making it specific to touch deprivation experienced during this period. For example, someone might be reporting little touch during COVID-19 but they might have been also experiencing little touch before, thus making it important to take these individual baselines into account. These findings are consistent with past research suggesting that when deprived of intimate touch, people show more mood and anxiety symptoms (48) and that those deprived of touch from close others report increased perceptions of loneliness (37). Interestingly, we also found a moderate to strong correlation between anxiety and loneliness (figure S1) and when accounting for loneliness on anxiety, our variable of interest (lack of intimate touch) no longer explains a statistically significant amount of variance. In contrast, our variable of

interest on loneliness remained statistically significant even after accounting for anxiety. Moreover, the latter model showed a higher R^2 (adjusted $R^2 = .22$) when including anxiety in the model vs. not, indicating that the regression model fits the observed data better, explaining 22% of the variance. This is not surprising given the tight relationship between touch and feelings of loneliness (10,37), but also between feelings of loneliness and anxiety (67-69). Taken together, these findings suggest that the effects of lack of intimate touch on loneliness go above and beyond the effects of touch on anxiety.”

p-values are sometimes reported as $p < .000$, and sometimes as $p = .000$. The p-value is never exactly .000. The authors should stick to the guidelines of the journal for reporting p-values. (APA-style is $p < .001$).

Response: We have now changed this all along the manuscript.

Figure 3. Please increase the font size for the numbers on the axes.

Response: We have now increased the font size, thank you.

Page 12, multilevel regression analysis. Please specify how lack of touch was controlled for and how the effect size (last line) was calculated.

Response: Lack of touch was controlled for by adding it as a covariate in our models, we have now specified this in the manuscript. Moreover, for consistency, we have now explained our effect sizes calculations in the methods, and included marginal and conditional R^2 instead of the original f^2 . These were computed using the `tab_model` function of the R package `sjPlot`. The marginal R-squared considers only the variance of the fixed effects, while the conditional R-squared takes both the fixed and random effects into account.

Page 8: “The following effect sizes for all analyses were computed using STATA : η_p^2 for repeated-measures ANOVA, r for correlations and η^2 for regressions. Marginal R^2 as well as Conditional R^2 were computed for the multilevel regressions using the `tab_model` function of the R package `sjPlot`. The marginal R-squared considers only the variance of the fixed effects, while the conditional R-squared takes both the fixed and random effects into account.”

Page 12, last paragraph, “touch by days in lockdown was significantly different”. Please provide the direction of the effect, also in other places.

Response: Thank you. We have now added this in the manuscript.

Figure S3. The crossed out correlation coefficients are a bit hard to read. Maybe crossing them out in grey would work better, otherwise try to mark them differently.

Response: We agree with the reviewer and took out the crossing, leaving blank the cells that were not significant, and coloring only significant correlations.

Reviewer: 2

Comments to the Author(s)

This study investigated how the restrictions imposed to contain the covid-19 pandemic affected experienced touch, touch deprivation and its effects on psychological well-being. In an online survey, participants were asked about their touch experiences during the restrictions, mental health and craving for touch. Moreover, the authors assessed whether attachment style and attitudes towards touch affected their touch experiences. A relatively large sample size of 1746 individuals participated. The results showed that friendly touch was lacking most during the restrictions and that intimate touch was experienced most but was also most wanted during the restriction. Importantly, the longer the duration of the restrictions, the more individuals craved intimate touch. This was also dependent on attachment style, with opposite patterns for anxiously-attached and avoidant-attached styles. Moreover, lack of intimate touch was associated with higher levels of anxiety, loneliness, worse level of mental health and less tolerance of isolation. Finally, the more positive individuals attitudes were towards touch, the more they wanted to experience friendly and intimate touch.

This study provides interesting and novel information about the effects of the covid-19 restrictions on touch experiences and its effects on mental health. Strong points of the review are its sample size and the relatively specific questions about the experienced touch during and before the restrictions. I do have a number of questions with respect to the questions used, the analyses and the interpretation:

We thank the reviewer for their positive and valuable feedback.

Methodology

- P. 7, 2.2.5: The way mental health was assessed seems very limited. Only 1 question was asked as a self-rating. Why not assess this with a more extensive standardized questionnaire? Also, while for loneliness and tolerance of isolation, the precise question is mentioned, this is not the case for mental health. Please add this question.

Response: Thank you for your remark. Indeed, given the possibility that single-items do not capture the domain of mental health concerns with enough granularity and precision, we instead focus our conclusions on our longer and validated measures such as anxiety and loneliness (please see our first point above). Only secondary, exploratory analyses on single items, as well as the precise question used to assess mental health, are presented only in the SM.

- Furthermore, to what extent do anxiety and mental health overlap?

Response: All our measures of well-being show a moderate to strong correlation (see Supplementary Figure S1). For the reasons explained above, we are now focusing on anxiety and loneliness, and given their correlation, we now account for each other in follow-up analyses (see response to reviewer 1 point 1). Interestingly, we found that when including loneliness in our model on anxiety, the lack of intimate touch does not

predict anxiety anymore. In contrast, when including anxiety in our model on loneliness, the lack of intimate touch still remains significant. Thus, our effects of intimate touch on loneliness remain significant even when accounting for anxiety. We have now included these analyses in the manuscript.

- P. 7, 2.2.7 Attitudes and Experiences towards touch: Which items exactly were selected from each component and why these items?

Response: We have now included the TEAQ selected items in the supplementary materials. These items were chosen because they had the highest loading of each of the 6 components of the Touch Attitudes and Experiences Questionnaire (TEAQ), yet 2 were selected from one of the components, namely the Child Touch experience, as they had the same loading (i.e., .80). We have now clarified this in the manuscript (p.8 and see also below).

In addition, we have now conducted confirmatory factor analyses. As expected, we found that the two items from ChT do indeed correspond to the same factor. Moreover, we found that each item belongs to a separate component or factor, consistent with the original paper validating the TEAQ (Trotter et al., 2018). We have now included the Table with loadings in Supplementary Materials (see also response to Reviewer 1 Comment 1) and our paragraph in the methods section now reads as follows:

p. 8:

“To examine attitudes and experiences towards touch, we used 7 items rated on a 5-point scale (1 = Disagree strongly and 7 = Agree strongly) from the Touch Experiences and Attitudes Questionnaire (TEAQ; (57)); see Supplementary Material for details on the selected items. Each item was selected as they correspond to one of the six components, and had the highest loading, from the TEAQ, namely: friends and family touch (FFT), current intimate touch (CIT), childhood touch (ChT), attitude to self-care (ASC), attitude to intimate touch (AIT) and attitude to unfamiliar touch (AUT). Note that two items from the childhood touch component were included as they both corresponded to the highest loading in the original scale (i.e., .80). Moreover, we conducted a factor analysis (using the `psych::fa` function of R; (62)) on these items, and found that the two items from ChT do indeed correspond to the same factor (see Supplementary Material Table S1). In addition, our factor analysis suggests that each item belongs to a separate factor or component, consistent with the original paper validating the TEAQ (57). After reverse-scoring appropriate items, items demonstrated moderate internal consistency, with Cronbach’s $\alpha = 0.63$. Items were summed to produce a total score for attitudes and experiences towards touch, with higher scores denoting more positive attitudes and experiences ($M=24.31$, $SD=5.27$). Averaging across items, touch attitudes and experiences score was $M=3.45$ ($SD=.76$).”

- Methods, p. 6: Has type of relationship with members of household been recorded for those not living alone? Do number of household members and type of a relation play a role in lack of touch and wanting touch?

Response: Thank you for this remark. We agree that the number of household members and type of relationship might play an interesting role on touch experience, and we have now added these analyses. However, please note that based on previous literature on the important of ‘quality’ rather than ‘quantity’ of relationship types and given the multiple, different combinations of concomitant relationships one may have in a household, we judged that the best way to investigate this question is to use both the quantity of household members and the subjective judgment of relational bonding instead of the ‘objective’ type of relationship that may not be subject to the same psychological influences. Specifically, we used the following question: “how close do you feel to the people living with you” (answered on a VAS from 0 to 100), which we think can be more informative than the type of relationship (e.g., mother, father, son, etc.) as the latter does not necessarily provide information about closeness (e.g., a person can live with their parents and yet have a distant relationship with them (particularly during lockdown) and consequently, less physical contact and less psychological comfort).

Regarding wanting touch, our analyses showed that the number of household members, the perceived closeness with them, and their interaction, does not influence the degree to which people crave or want touch during COVID.

Regarding touch experience, our analyses showed that participants reported more touch experienced before than during COVID-19; that the higher the perceived closeness with the people they live with, the higher the amount of touch experienced; and that individuals report experiencing less touch when living with 6 or more people. However, for individuals living with 6 people or more: the closer they feel with the people in their household, the more the amount of touch they report to have experienced *before* COVID19 (see figure below and supplementary materials for details).

We have now included these analyses in supplementary materials and make reference to them in the methods section of the paper as follows:

p. 10 *“... The number of household members or how close they feel with them does not influence the amount of touch people would have wanted to experience in the past week i.e., during COVID19. However, the closer they feel with the people they live with, the higher the amount of touch they report to have experienced, irrespective of before or during COVID-19. Interestingly this is particularly the case for people living with more than 5 people before but not during COVID-19 related restrictions (see supplementary materials Figure S2).”*

- Why use a 5-point scale for some questions (loneliness, mental health) while for others a VAS from 0-100 was used (tolerance to isolation)?

Response: We have now moved single-item questions to SM (see rationale above). Our analyses now focus on anxiety and loneliness and both have either a 4-point or 5-point scale, according to their questionnaire guideline.

- Similarly, it is not entirely clear how practicing social distancing was measured. Did the participants rate this on a VAS from 1-100?

Response: Indeed, this question was answered on a VAS scale, ranging from 1 to 100. We have now added more details on all questions and scales used in the Methods section.

Results

Main analyses, 3.2.1, pp 10-11: the R² for all regression analyses, while significant, are relatively low (.02 for all except loneliness). Thus only a small percentage of the variance can be explained by experienced touch. Perhaps this could be discussed in the Discussion

Response: We thank the reviewer for this remark.

We have now addressed this point in the discussion (not only in relation to the other psychological variables but also an observed higher R² when we include anxiety in our regression model on loneliness). It now reads as follows:

P. 16-17. *“ COVID-19-related restrictions inevitably affected core social habits of citizens, including tactile behaviours (with our data supporting this notion; see Figure 1 and Figure 2). Given growing lab and epidemiological evidence suggesting that social touch has beneficial effects on well-being (12,33,34), the present study first investigated whether the touch deprivation caused by COVID-19-related restrictions was associated with worse psychological outcomes. We found that the more the intimate touch (but not friendly or professional) experienced in the past week (i.e., during COVID-19), the better the targeted psychological outcomes: self-reported anxiety and feelings of loneliness (see Supplementary Material for similar exploratory findings on single items measuring mental health and tolerating isolation), with the magnitude of this effect*

being small ($\eta^2=.01$) and moderate to large ($\eta^2=.09$), respectively. These findings are consistent with growing evidence suggesting that the beneficial effects of touch are context-specific (6,39). Indeed, touch is central to intimate, romantic relationships (66), and the regulatory role of touch seems to be mediated by psychological intimacy (51). These findings are important given that anxiety, depression and stress have been shown to be common reactions to the COVID-19 pandemic (50) and intimate touch may work as a protective factor. Interestingly though, while significant, the R^2 was low for anxiety (and mental health, tolerating isolation) although less so for loneliness (i.e., $R^2 =.09$; see Supplementary materials), indicating that the latter is a better fit for the model yet only a small percentage of the variance can be explained by experienced touch. Given evidence suggesting that certain experiences that are likely to be experienced during COVID-19 may predict worse mental health outcomes (e.g., low income predicts mental distress (67) and illnesses or death of a close other predicts loneliness (68)), it is possible that other factors that determine anxiety and loneliness, that were not tested here (e.g., self-isolation history, conditions of work and income during lockdown) play a critical role in explaining variance.

Moreover, we found that the more the lack of intimate touch (but not friendly or professional), the worse the self-reported anxiety and feelings of loneliness. Importantly, unlike the above findings on touch experienced in the past week, lack of touch computations take into account touch experienced before COVID-19 (i.e., baseline), thus making it specific to touch deprivation experienced during this period. For example, someone might be reporting little touch during COVID-19 but they might have been also experiencing little touch before, thus making it important to take these individual baselines into account. These findings are consistent with past research suggesting that when deprived of intimate touch, people show more mood and anxiety symptoms (48) and that those deprived of touch from close others report increased perceptions of loneliness (37). Interestingly, we also found a moderate to strong correlation between anxiety and loneliness (figure S1) and when accounting for loneliness on anxiety, our variable of interest (lack of intimate touch) no longer explains a statistically significant amount of variance. In contrast, our variable of interest on loneliness remained statistically significant even after accounting for anxiety. Moreover, the latter model showed a higher R^2 (adjusted $R^2 =.22$) when including anxiety in the model vs. not, indicating that the regression model fits the observed data better, explaining 22% of the variance. This is not surprising given the tight relationship between touch and feelings of loneliness (10,37), but also between feelings of loneliness and anxiety (67-69). Taken together, these findings suggest that the effects of lack of intimate touch on loneliness go above and beyond the effects of touch on anxiety.”

Minor comments

- P. 6, line 12: “368 within Europe” please change to “368 within continental Europe”

Response: We have now changed this.

- Conclusion, p. 18: “social distancing” is mentioned, however according to the introduction this should be “physical distancing”.

Response: We thank the reviewer for spotting this, we have now changed it to “in times of physical distancing”.

- Supplementary material contains a Figure S2 and a Table S6 which reported analyses concerning psychiatric conditions. These are only referred to in the Discussion of the manuscript, without mentioning the Figure and Table numbers. Please add those.

Response: We now added the reference to the supplementary figure and table in the discussion.

Appendix C

The authors have done a great job in revising the article and in thoroughly responding to each of the issues raised. The revised structure has also contributed to a much better readability. Due to these changes, the manuscript is much improved. I am convinced it will reach and interest a wide audience.

I still have some minor points and details that I ask the authors to complete in order to further increase the manuscript's clarity and impact. In particular, the confound of wish for touch and amount of touch received which is inherent in the applied measure of the amount of touch needs to be discussed more carefully.

Please address the following points (listed in the order of their appearance in the manuscript):

Abstract, line 4 “is associated with worse psychological wellbeing”. Please replace “worse psychological wellbeing” with something like “higher anxiety and greater loneliness”, as these constructs were the focus of this revision.

Page 6, section 2.2.2. Thanks for providing a screenshot of the items in the form they were presented to the participant. This nicely illustrates the question format at a glance, and I think the readers would appreciate it as well. So I ask the authors to include it also in the paper.

Page 7, questionnaires on anxiety, loneliness, ECR-S and attitudes and experiences towards touch. Please name the maximum and minimum possible total score for each questionnaire. For the ECR, it is clear that the smallest score (mean score for the whole group) that is possible to obtain is 0 and the maximum score is 7, but for the STAI-SF this is not immediately clear. Knowing the minimum and maximum scores possible allows the reader to immediately grasp whether the obtained group score indicates a value high or low on the trait.

Page 8, factor analysis for the 7 items from the TEAQ. The authors state that they found that each item belongs to a separate factor, but it is not clear how they arrived at that conclusion. It appears that what was done was an exploratory factor analysis – was the number of factors determined based on eigenvalues, on a Scree plot, or any other method? Related to this, it is not quite clear which values table S1 shows - these cannot be eigenvalues? Also, what is “h2”, “u2”, and “com”? Please complete.

Page 8, line 24 (section “descriptive statistics and preliminary analyses”). Explain how the interactions were resolved.

Page 9, section B3. I may have missed something, but I wonder why the authors calculate three different regression analyses, all with the same outcome variable? All the predictors are personality variables, and it would add information to add all these in one multiple linear regression analysis. For example, accounting for the variance explained by the 7-item TEAQ when looking at the relationship between attachment anxiety and wanting may reduce the variation and give estimates that are more precise.

Regarding the analysis with every single item described in the same section, lines 38-41 – was multicollinearity checked here as well? One would intuitively expect a high correlation between these items.

Although loneliness in this manuscript is understood as a consequence, it would be highly interesting to include loneliness as a predictor variable in the same regression. It is well possible

that higher loneliness to begin with explains higher wanting of touch, irrespective of any social distancing regulations. Indeed, your analysis of anxiety hints at the powerful role of loneliness in this context. This analysis may be too much for this paper (you decide), but I believe would be important to explore.

Page 10, lines 1-4. It is a bit unclear what was tested here. I assume that before Covid-19 versus past week was tested separately for friendly touch, intimate and professional touch? And each of these comparisons was significant with $p < .001$? Please make this clearer, otherwise, one could be tempted to think that what was compared was the difference between friendly touch versus intimate and professional touch.

Page 10, lines 11-12 “note that the same pattern of effects remains when applying Bonferroni correction”? I assume that the Bonferroni-correction does not refer to the Anova, but to potential post-hoc tests performed to resolve the interaction? It is not clear why an Anova would need an additional Bonferroni-correction. Please specify in the methods.

Figure 1. I would prefer to add the word “subjective” to the y-axis and call this variable “subjective amount of touch”, and remove it from the figure caption. This would be more precise. There are no violins for Figure B?

Page 11, line 40. The word “touch” is missing.

Page 11, paragraph 3.1.3. and following: please add the N (number of participants/data points in this analysis) to the results from the correlation analyses, e.g., $r(N) = .09$, $p < .001$).

Page 12. “when including loneliness in the model, the lack of intimate touch is no longer significant”. In other words – the lack of touch may not be a problem unless one is lonely as well. Such an important finding! Still, the authors chose to only mention this in one sentence in the discussion. I leave this up to the authors, but think this finding would deserve much more space as it has implications beyond the current pandemic effects on touch research in general. For example, it shows the importance of considering loneliness when designing studies on touch effects.

Figure 3. The pink and red lines look very similar in the printed version of the document. Why not use black, for example?

Page 19, middle paragraph. The argumentation is a bit messy, but then I see I was also not entirely clear in my comment. The authors write that the amount of touch measured is likely influenced by longing for touch, “similarly to other studies of touch”. However, this problem is particularly present in the current study. The measure used by Bessler et al also has its issues (one may, for example, question how accurately participants can remember each occasion of casual touch during an entire week). Still, as it is an absolute measure of touch frequency (i.e., numerical frequencies), the values are comparable. Someone that reports 10 hugs remembers twice as many hugs than someone who reports 5 hugs.

Verbal labels for frequencies are more vague and ambiguous (e.g., Nakao and Axelrod 1983), and there is a body of literature the advantages and disadvantages of using one or the other in questionnaires, and how to translate verbal labels to numerical frequencies (e.g. Bocklisch et al. 2012). With the verbal labels (“a lot” and “not at all”) used in the present study, comparisons between individuals are not possible, as “a lot” can mean largely different things for two persons. Even more, and this is the more problematic issue, the wish for touch is intertwined

with the frequency response. People with a high wish for touch who have received a high number of touch, but still would want more, will presumably report the same value as people who have a low wish for touch and have received very little touch. Thus, a response close to “not at all” can be given by both persons that have received a lot or almost no touch. It is in principle impossible to distinguish the wish for touch from the amount of touch received in the responses provided. The authors argue that this is not a problem because they calculated a difference value, but also here a large difference could indicate all of the following: 1) a high wish for touch in the last week, but a lower wish for touch before the pandemic, with the same frequency of touch, 2) a high wish for touch in the last week and before the pandemic, and a lower frequency of touch, 3) a low wish for touch in the last week and a low wish for touch before the pandemic, together with a lower frequency of touch. All in all, I ask the authors to present and discuss the differences between this verbal measure and the numerical measures of frequencies used by other authors, both pros and cons. For example, Sailer and Ackerley (2017) used a combination of numerical and verbal frequencies, but unfortunately, without analysing the relationship between the two. Please also address the potential impact of the confound of frequency and touch wish with greater care.

Line 27, same page “supportive of this notion” does not make sense. These findings do not support the notion that longing and amount of touch are related. I mentioned these findings because they may fit to the results of the current paper where participants did not crave professional touch (assuming that handshakes with individuals other than the partner can be counted as professional touch). Only include this if you think it is relevant.

Same paragraph, last sentence. Add reference [15], because this is exactly what these authors did (EMA and additionally, observation).

References:

Bocklisch F, Bocklisch SF, Krems JF (2012). Sometimes, often, and always: Exploring the vague meanings of frequency expressions. *Behavior Research Methods*. 44:144–157.

Nakao MA, Axelrod S. Numbers are better than words. Verbal specifications of frequency have no place in medicine (1983). *Am J Med*. 74(6):1061-5

Appendix D

Associate Editor Comments to Author (Dr Rochelle Ackerley):

I agree with the reviewers that the authors have done an excellent job of revising their manuscript and answering the comments. I am recommending that your paper is accepted, but some final minor revisions are required (moreso from Reviewer 1). Reviewer 2 also states that the datafile is available and easy to understand, but they cannot find any data analyses syntax files, therefore could these please be made additionally available?

Response: We thank the Associate Editor. We have now addressed the final minor revisions from reviewer 1 and 2, and have now put on OSF the data analyses syntax files.

Reviewer: 1

Comments to the Author(s)

The authors have done a great job in revising the article and in thoroughly responding to each of the issues raised. The revised structure has also contributed to a much better readability. Due to these changes, the manuscript is much improved. I am convinced it will reach and interest a wide audience.

I still have some minor points and details that I ask the authors to complete in order to further increase the manuscript's clarity and impact. In particular, the confound of wish for touch and amount of touch received which is inherent in the applied measure of the amount of touch needs to be discussed more carefully.

Response: We thank the reviewer for their valuable feedback. Please see below our responses, with changes to the manuscript in colour blue.

Please address the following points (listed in the order of their appearance in the manuscript):

Abstract, line 4 "is associated with worse psychological wellbeing". Please replace "worse psychological wellbeing" with something like "higher anxiety and greater loneliness", as these constructs were the focus of this revision.

Response: Thank you, we have now replaced this.

Page 6, section 2.2.2. Thanks for providing a screenshot of the items in the form they were presented to the participant. This nicely illustrates the question format at a glance, and I think the readers would appreciate it as well. So I ask the authors to include it also in the paper.

Response : As suggested, we have now included this as a Figure in the manuscript.

Figure 1. Example of the visual analogue scale ranging from 0 'not at all' to 100 'a lot' for three items corresponding to different types of 'social' touch. The cursor was initially placed at 0, and participants had to move the cursor in order for the question to be validated. In this illustrative example, participants are asked about the amount of touch experienced in the past week (i.e., during COVID-19). However, participants were also asked about their tactile experience in relation to the amount of touch experienced before COVID-19 as well as their wanting to have experienced these types of touch in the past week.

Page 7, questionnaires on anxiety, loneliness, ECR-S and attitudes and experiences towards touch. Please name the maximum and minimum possible total score for each questionnaire. For the ECR, it is clear that the smallest score (mean score for the whole group) that is possible to obtain is 0 and the maximum score is 7, but for the STAI-SF this is not immediately clear. Knowing the minimum and maximum scores possible allows the reader to immediately grasp whether the obtained group score indicates a value high or low on the trait

Response: We have now included the minimum and maximum scores possible for anxiety, loneliness, ECR-S and TEAQ.

Page 8, factor analysis for the 7 items from the TEAQ. The authors state that they found that each items belongs to a separate factor, but it is not clear how they arrived at that conclusion. It appears that what was done was an exploratory factor analysis – was the number of factors determined based on eigenvalues, on a Scree plot, or any other method? Related to this, it is not quite clear which values table S1 shows - these cannot be eigenvalues? Also, what is "h2", "u2", and "com"? Please complete.

Response: We thank the reviewer for their point, as it was not really clear. We have now clarified our analysis and table S1.

Actually, we performed a confirmatory factor analysis, not exploratory, as we took items from different factors of the full TEAQ. We took items from 6 factors of TEAQ, and we wanted to confirm that our 7 items factor into the same 6 factors as expected. We confirmed that two of the items belong to the same factor (PA1) and the rest are one factor per item. This is what we expected as these two items were initially taken from the same original factor of the full TEAQ (“Childhood touch”).

Table S1 shows the “loadings” of each item in each of the 6 factors. We determine which item belongs to which factor by identifying in which factor it has its higher loading, and also checking that the loading is higher than the commonly accepted minimum threshold of 0.4 and that the complexity (*com*) of the item is as close as possible to 1 (i.e. that it loads to only one factor). In our table e.g. TEAQ_parents and TEAQ_tuck both have their highest loading (0.75 and 0.73) in factor PA1 and both have a complexity very close to 1 (*com*=1.1). Therefore, we can determine they belong to the same factor, as expected, as they were items taken from the same factor of the “full” TEAQ. Similarly, TEAQ_bath has its highest loading (1) in factor PA2 with complexity 1. And so on.

Moreover, **h2** corresponds to the communality estimates for each item. These are merely the sum of squared factor loadings for that item. The communality for a variable is the amount of variance accounted for by all of the factors. That is to say, for orthogonal factors, it is the sum of the squared factor loadings (row-wised).

u2 corresponds to the uniqueness (equal to 1-**h2**). Uniqueness is the variance that is ‘unique’ to the variable and not shared with other variables. Notice that the greater ‘uniqueness’ the lower the relevance of the variable in the factor model.

Com corresponds to the Hoffman's index of complexity for each item. This is $\{(\sum a_i^2)^2 / \sum a_i^4\}$ where a_i is the factor loading on the i^{th} factor. From Hofmann (1978), MBR. See also Pettersson and Turkheimer (2010). It tells how much an item reflects a single construct. It equals one if an item loads only on one factor, 2 if evenly loads on two factors, etc.

Page 8, line 24 (section “descriptive statistics and preliminary analyses”). Explain how the interactions were resolved.

Response: Thank you, we have now added that (p.9).

Page 9, section B3. I may have missed something, but I wonder why the authors calculate three different regression analyses, all with the same outcome variable? All the predictors are personality variables, and it would add information to add all these in one multiple linear regression analysis. For example, accounting for the variance explained by the 7-item TEAQ when looking at the relationship between attachment anxiety and wanting may reduce the variation and give estimates that are more precise.

Response:

We agree with the reviewer and have now merged the regression model 1 and 2, as the predictor variables (avoidance score, anxiety score, TEAQ total score) are the total scores of each questionnaire (i.e., same dimensionality). Note that we obtain the exact same pattern of results, except for the directionality of the TEAQ by type of touch interaction (i.e., only friendly touch driving the interaction). This has now been changed throughout (including

Figures 6 and 7). However, we left the analyses on the subcomponents of the TEAQ as a separate analysis as these add another dimension and we believe should not be included as part of the same model looking at the total score of the same personality variable. Moreover, in this follow up model we were not interested in their interaction with the different types of touch. This is specified in page 10 as follows: *“To further examine whether wanting touch was associated with specific components of the attitudes and experiences towards touch measure, we specified a multilevel regression model with wanting touch ratings as the outcome variable and included the seven items of the experiences and attitudes towards touch measure as predictor variables”*.

Regarding the analysis with every single item described in the same section, lines 38-41 – was multicollinearity checked here as well? One would intuitively expect a high correlation between these items.

Response: Indeed, one would expect a high correlation between some of these items, but (i) these were mean-centred which minimizes multi-collinearity issues, moreover (ii) their overall VIF was of 1.56 which suggests no problematic amount of multicollinearity in our model including these items.

Although loneliness in this manuscript is understood as a consequence, it would be highly interesting to include loneliness as a predictor variable in the same regression. It is well possible that higher loneliness to begin with explains higher wanting of touch, irrespective of any social distancing regulations. Indeed, your analysis of anxiety hints at the powerful role of loneliness in this context. This analysis may be too much for this paper (you decide), but I believe would be important to explore.

Response: Indeed, one could explore the impact of loneliness on wanting to experience touch, even though, as the reviewer mentions, it is possible that this effect is irrespective of any social distancing regulations. Along this point, we have explored the relationship between loneliness and wanting to experience touch in the past week (as an average). We find that greater loneliness predicts more wanting touch in the past week (see below). We include the analysis here for the reviewer’s knowledge, but we agree this analysis is too much for the current paper.

```
. regress average_missingtouch mean_loneliness
```

Source	SS	df	MS	Number of obs	=	1,490
Model	46930.0525	1	46930.0525	F(1, 1488)	=	71.50
Residual	976604.918	1,488	656.320509	Prob > F	=	0.0000
Total	1023534.97	1,489	687.397562	R-squared	=	0.0459
				Adj R-squared	=	0.0452
				Root MSE	=	25.619

average_missi~h	Coef.	Std. Err.	t	P> t	[95% Conf. Interval]
mean_loneliness	3.458089	.4089485	8.46	0.000	2.655912 4.260266
_cons	35.5409	1.618653	21.96	0.000	32.36581 38.71598

Page 10, lines 1-4. It is a bit unclear what was tested here. I assume that before Covid-19 versus past week was tested separately for friendly touch, intimate and professional touch? And each of these comparisons was significant with $p < .001$. Please make this clearer, otherwise, one could be tempted to think that what was compared was the difference between friendly touch versus intimate and professional touch.

Response: We first report main effects from the ANOVA (i.e., time and type of touch), that is why at first we are comparing time (before vs after), irrespective of type of touch, and then type of touch, irrespective of time. We have now clarified this and added those specific comparisons when reporting the time by type of touch interaction (and not just what was driving the interaction). We have also made that sentence clearer.

Page 10: “As presented in Figure 1, participants reported more touch experienced before COVID19 ($M=51.59$, $SD=25.79$) as compared to the amount of touch reported in the past week ($M=16.68$, $SD=15.06$), $F(1, 1489) = 3306$, $p < .001$, $\eta_p^2 = .69$, *irrespective of type of touch*. Intimate touch ($M=52.18$, $SD=32.67$) was reported as the most experienced, as compared to friendly ($M=28.84$, $SD=18.09$) and professional ($M =21.38$, $SD=17.51$) touch, $F(1.4, 2084.7) = 1018.2$, $p < .001$, $\eta_p^2 = .41$, *irrespective of time*. The type of touch interacted with time, $F(1.763, 2625.1) = 473.3$, $p < .001$, $\eta_p^2 = .24$. Touch was reported significantly less in the past week (vs. before COVID-19) separately for intimate, friendly and professional touch, p 's $< .001$. The type of touch by time interaction was driven by a larger difference between touch in the past week vs. before COVID-19 in friendly ($M=47.55$, $SD=31.94$), as compared to intimate ($M=19.85$, $SD=32.10$) and professional touch ($M=37.33$, $SD=32.06$), p 's $< .001$.”

Page 10, lines 11-12 “note that the same pattern of effects remains when applying Bonferroni correction”? I assume that the Bonferroni-correction does not refer to the Anova, but to potential post-hoc tests performed to resolve the interaction? It is not clear why an Anova would need an additional Bonferroni-correction. Please specify in the methods

Response: Thank you, we have now clarified it is related to post-hoc tests.

Figure 1. I would prefer to add the word “subjective” to the y-axis and call this variable “subjective amount of touch”, and remove it from the figure caption. This would be more precise. There are no violins for Figure B?

Response: As suggested by the reviewer we now added the word subjective to the y-axis. Violins for friendly and professional in Figure B are very small and close to zero, this makes sense as this corresponds to the types of touch that we were expecting to be almost non-existent during lockdown (and as such skewed). Indeed, this was also another reason for doing our main analyses on ‘lack of touch’. While the violin for intimate touch is almost totally flat (in comparison to the two others) as it is equally distributed between the two extremities (i.e. people having a lot vs. no intimate touch).

Page 11, line 40. The word “touch” is missing.

Response: We apologize but we cannot spot the word missing in the figure’s legend “(C) “In the past week, How much would you have wanted to experience these different types of 'social' touch?” from 0=not at all, to 100=a lot.” We are guessing that it is because (C) question is not mirroring the (A) and (B) question where the word “touch” is coming after “How much”.

Page 11, paragraph 3.1.3. and following: please add the N (number of participants/data points in this analysis) to the results from the correlation analyses, e.g., $r(N)=.09$, p

Response: Thank you, we have now added this.

Page 12. “when including loneliness in the model, the lack of intimate touch is no longer significant”. In other words – the lack of touch may not be a problem unless one is lonely as well. Such an important finding! Still, the authors chose to only mention this in one sentence in the discussion. I leave this up to the authors, but think this finding would deserve much more space as it has implications beyond the current pandemic effects on touch research in general. For example, it shows the importance of considering loneliness when designing studies on touch effects.

Response: Thank you very much for pointing this out, we have now expanded on this as follows:

P. 17: “Moreover, we found that the more the lack of intimate touch (but not friendly or professional), the worse the self-reported anxiety and feelings of loneliness. Importantly, unlike the above findings on touch experienced in the past week, lack of touch computations take into account touch experienced before COVID-19 (i.e., baseline), thus making it specific to touch deprivation experienced during this period. For example, someone might be reporting little touch during COVID-19 but they might have been also experiencing little touch before, thus making it important to take these individual baselines into account. These findings are consistent with past research suggesting that when deprived of intimate touch, people show

more mood and anxiety symptoms (48) and that those deprived of touch from close others report increased perceptions of loneliness (37). Interestingly, we also found a moderate to strong correlation between anxiety and loneliness (Figure S1) and when accounting for the effect of loneliness on anxiety, our variable of interest (lack of intimate touch) no longer explains a statistically significant amount of variance. *In other words, touch deprivation may not be a problem, at least for anxiety, unless one is lonely as well. Such finding has implications for touch research beyond the current pandemic effects. For example, future research on touch should consider loneliness when designing studies on touch effects. In contrast, when accounting for the effect of anxiety on loneliness, our variable of interest (lack of intimate touch) remained statistically significant. This suggest that lack of touch during social distancing had effects on feelings of loneliness, even when controlling for related feelings of anxiety. Moreover, the latter model showed a higher R² (adjusted R² =.22) when including anxiety in the model vs. not, indicating that the regression model fits the observed data better, explaining 22% of the variance. This is not surprising given the tight relationship between touch and feelings of loneliness (10,37), but also between feelings of loneliness and anxiety (e.g., (69–71)). Taken together, these findings suggest that the effects of lack of intimate touch on loneliness go above and beyond the effects of touch on anxiety”*

Figure 3. The pink and red lines look very similar in the printed version of the document. Why not use black, for example?

Response: We have now changed the pink line to maroon (same for the data points).

Page 19, middle paragraph. The argumentation is a bit messy, but then I see I was also not entirely clear in my comment. The authors write that the amount of touch measured is likely influenced by longing for touch, “similarly to other studies of touch”. However, this problem is particularly present in the current study. The measure used by Bessler et al also has its issues (one may, for example, question how accurately participants can remember each occasion of casual touch during an entire week). Still, as it is an absolute measure of touch frequency (i.e., numerical frequencies), the values are comparable. Someone that reports 10 hugs remembers twice as many hugs than someone who reports 5 hugs.

Verbal labels for frequencies are more vague and ambiguous (e.g., Nakao and Axelrod 1983), and there is a body of literature the advantages and disadvantages of using one or the other in questionnaires, and how to translate verbal labels to numerical frequencies (e.g. Bocklisch et al. 2012). With the verbal labels (“a lot” and “not at all”) used in the present study, comparisons between individuals are not possible, as “a lot” can mean largely different things for two persons. Even more, and this is the more problematic issue, the wish for touch is intertwined with the frequency response. People with a high wish for touch who have received a high number of touch, but still would want more, will presumably report the same value as people who have a low wish for touch and have received very little touch. Thus, a response close to “not at all” can be given by both persons that have received a lot or almost no touch. It is in principle impossible to distinguish the wish for touch from the amount of touch received in the responses provided. The authors argue that this is not a problem because they calculated a difference value, but also here a large difference could indicate all of the following: 1) a high wish for touch in the last week, but a lower wish for touch before the pandemic, with the same frequency of touch, 2) a high wish for touch in the last week and before the pandemic, and a lower frequency of touch, 3) a low wish for touch in the last week and a low wish for touch before the pandemic, together with a lower frequency of touch. All in all, I ask the authors to present and discuss the differences between this verbal measure and the numerical measures of frequencies used by other authors, both pros and cons. For example, Sailer and Ackerley (2017) used a combination of numerical and verbal frequencies, but unfortunately, without analysing the relationship between the two. Please also address the potential impact of the confound of frequency and touch wish with greater care.

Line 27, same page “supportive of this notion” does not make sense. These findings do not support the notion that longing and amount of touch are related. I mentioned these findings because they may fit to the results of the current paper where participants did not crave professional touch (assuming that handshakes with individuals other than the partner can be counted as professional touch). Only include this if you think it is relevant.

Same paragraph, last sentence. Add reference [15], because this is exactly what these authors did (EMA and additionally, observation).

Response: Thank you very much for this clarification. This paragraph now reads as follows:

P. 19: "Second, similarly to other studies on touch, the amount of touch reported to have been experienced in the past week is likely influenced by the longing for touch (as can also be observed in Figure 2), with such problem being particularly present in the current study as it relied on verbal labels for frequencies, which are more vague and ambiguous (79). Specifically, our study relied on retrospective estimates of experience that may be influenced both by experience itself and one's own reflective biases, including the desire for touch. Even though we partly controlled for some of these biases by also measuring the amount of touch experienced before COVID-19 (subject also to reflective biases), comparisons between individuals becomes problematic as "a lot" can mean largely different things between individuals. Moreover, the fact that the longing for touch may be intertwined with the frequency of response makes it difficult to distinguish the wish for touch from the amount of touch received in the responses provided. For example, people with a high wish for touch who have received a high number of touch, but still would want more, will presumably report the same value as people who have a low wish for touch and have received very little touch. Thus, other measures of touch experience are needed to account for the possibility of these measures (estimates of experienced touch and one's desire for touch) not being independent. For example, other studies have relied on asking for the absolute frequency of touch (i.e., a guessed count (15,31,74)). As an absolute measure of touch frequency (i.e., numerical), the values are comparable, although one may still question how accurately participants can remember each occasion of casual touch during an entire week. Moreover, some studies have for example used a combination of numerical and verbal frequencies (77) but without analysing the relationship between the two. Alternatively, studies could rely on ambulatory assessment methods that best capture ongoing experience and not retrospective reflections on one's experience (15)."

Reviewer: 2

Comments to the Author(s)

Overall, the authors have done a very good job revising the manuscript and have addressed my previous concerns. I have one remaining request: Figures 4, 5 and 6 are very difficult to read as the lines for the different types of touch are more or less obscured by the individual data points. It would be nice if this could be improved. Also for Figure 2, the scale of the y-axis is much smaller than that of the x-axis, which makes the significant relationships more difficult to detect.

Response: We thank the reviewer for their positive feedback. Figures 4, 5 and 6 (now 5, 6 and 7): We have now reduced the size of the individual data points and reduced their opacity. As suggested, for Figure 2 (now Figure 3) we have also made the scale of the y-axis bigger.

Figure 3 (old figure 2)

Figure 5 (old figure 6)

Figure 6 (old figure 5)

Figure 7 (old figure 6)